# Predicting Gene Expression in Spatially Resolved Transcriptomics Across Samples Through Probabilistic Fusion of Hierarchical Histology and Spatial Information

## Abstract

Spatially resolved transcriptomics (SRT) is a transformative technology in biomedical research, yet its scalability is hindered by high costs and restricted capture areas. Computational methods for predicting high-quality gene expression are needed. However, existing methods are ineffective at predicting high-dimensional gene expression and generalizing to multiple spatial slices, primarily due to inter-sample heterogeneity and ineffective integration of visual and spatial information. To address these challenges, we propose STevs, a deep generative model designed to predict gene expression from tissue histology through a probabilistic fusion of image and spatial representations. STevs employs a multimodal variational autoencoder (VAE) architecture featuring parallel encoders that process distinct modalities: a Swin Transformer for hierarchical visual representation extraction and a multilayer perceptron (MLP) for spatial coordinates. The latent representations from these modalities are fused under uncertainty using a Product of Experts (PoE) mechanism. Furthermore, we introduce a latent alignment loss to explicitly promote a shared representation across modalities, thereby ensuring consistency between the image and spatial latent spaces. Comprehensive experimental evaluations demonstrate that STevs not only achieves state-of-the-art performance on standard within-slice gene prediction tasks but also significantly outperforms existing methods in the more challenging cross-slice prediction scenario. Our work provides a powerful computational tool capable of predicting gene expression directly from histology images, reducing the need for costly SRT experiments.

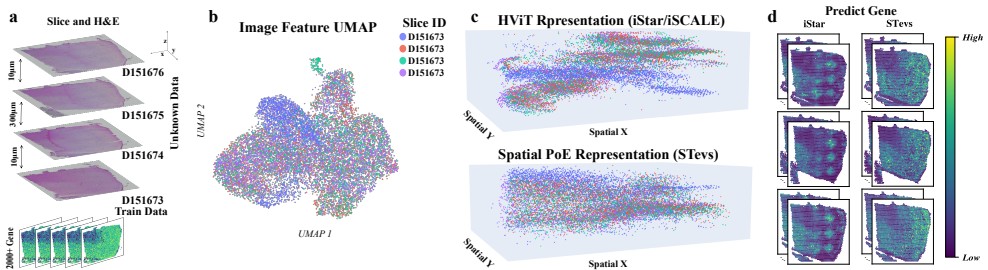

Figure 1: Workflow and representation space visualization. (a) The training and testing datasets. (b) Uniform Manifold Approximation and Projection (UMAP) McInnes et al. (2018) visualization of HViT-encoded features. (c) Spatial distribution of HViT representation across samples before and after PoE fusion. Before fusion, HViT features show clear separation between samples (upper panel). After PoE fusion, the features exhibit a more blended distribution, demonstrating improved cross-sample integration (lower panel). (d) An example of spatial gene prediction based on two different representation.

# 1 INTRODUCTION

Understanding the spatial organization of cells and gene expression patterns within tissues is essential for uncovering fundamental biological processes in fields such as developmental biologyAsp et al. (2019); Cui et al. (2023), neuroscienceChen et al. (2020), and cancer researchJi et al. (2020); Moncada et al. (2020). Recent advances in spatially resolved transcriptomics (SRT) technologies — including 10X Visium Andersson et al. (2021), Slide-seq V2 Stickels et al. (2021), and Stereo-seq Wang et al. (2022), have enabled whole-transcriptome profiling while preserving spatial context Burgess (2019); Rao et al. (2021). These technologies offer unprecedented opportunities to construct detailed tissue atlases, decipher cell–cell interactions, and explore the tumor microenvironment Williams et al. (2022); Miao et al. (2024).

However, the high experimental cost and limited capture area of SRT experiments (e.g., only 6.5 × 6.5 mm capture area of 10X Visium) hinder their broad application in clinical samples and large-scale cohort studies Schmauch et al. (2020); Gao et al. (2024). In contrast, hematoxylin and eosin (H&E)-stained histology images, the gold standard in pathological diagnosis, are widely available, cost-effective, and rich in cellular and tissue structural information Yu et al. (2016). Growing evidence indicates a strong correlation between tissue histology and gene expression patterns Naik et al. (2020); Wagner et al. (2023), suggesting the feasibility of predicting spatial gene expression directly from H&E images Long et al. (2023). This premise has motivated the development of numerous computational models. The technical evolution has progressed from initial convolutional neural networks (CNNs) processing individual image patches He et al. (2020); Monjo et al. (2022), to graph neural networks (GNNs) characterizing spatial contextual relationships Hu et al. (2021); Zeng et al. (2022); Gao et al. (2024), and more recently to vision transformers that capture long-range and hierarchical tissue features Pang et al. (2021); Zhang et al. (2024); Chung et al. (2024).

Despite these advances, existing methods face three major challenges: (i) Limited generalization ability: Models typically perform well on their training tissue slides but suffer significant performance degradation when applied to new slides from different individuals or batches (Fig. 1a), even in the absence of apparent image batch effects (Fig. 1b). This performance degradation stems primarily from shifts in image features that persist even at spatially adjacent locations across different tissue slides (Fig. 1c)Andersson et al. (2021); Pang et al. (2021). (ii) Poor scalability to high-dimensional gene expression: Most methods are designed for low-dimensional gene targets (typically < 1000 genes) He et al. (2020); Pang et al. (2021); Chung et al. (2024); Yang et al. (2024), necessitating the exclusion of substantial gene information from the full transcriptomics data. (iii) ineffective multimodal integration: Current approaches predominantly rely on simplistic integration strategies like feature concatenation or graph message passing, which fail to capture the complex interdependencies between gene expression, cellular histology, and spatial information Anderson & Simon (2020), and consequently lacking the ability to robustly model uncertainty across heterogeneous information sources Baltrušaitis et al. (2018).

To address the aforementioned challenges, we propose STevs (Spatial Transcriptomics gene expression prediction by integrating visual representations and spatial information), a novel deep generative framework that robustly predicts spatial gene expression by probabilistically integrating hierarchical visual histology with spatial information (Fig. 2a). Our objective is to learn an intrinsic and generalizable mapping from histology to high-dimensional gene expression that transfers effectively across tissue slides (Fig. 1d). The key contributions of this work are as follows: First, we designed a VAE framework Kingma & Welling (2013); Suzuki et al. (2016) that utilizes parallel encoders to learn the hierarchical visual features of tissue images and the contextual information of spatial coordinates, respectively. The model incorporates a negative binomial (NB)-based decoder Lopez et al. (2018) to directly characterize discrete and over-dispersed SRT count data, enabling accurate prediction of high-dimensional gene expression. Second, we incorporated a Product of Experts (PoE) mechanism Hinton (2002) to probabilistically fuse the latent distributions from the visual and spatial modalities, thereby obtaining a more robust joint representation that accounts for uncertainty (Fig. 1c). Finally, we proposed a latent space alignment loss Ji et al. (2020); Wagner et al. (2023) that enhances unified representation learning by explicitly constraining cross-modal latent spaces, ensuring consistency and mutual information exchange between modalities. Extensive experiments on 16 datasets across 5 groups demonstrate that STevs achieves state-of-the-art performance on standard intra-slice prediction tasks and significantly outperforms existing advanced methods in the more challenging cross-slice prediction task (Fig. 2c). Our work pro-

vides a reliable and generalizable solution for generating high-quality virtual spatial transcriptomics data, paving a promising path for advancing large-scale molecular analysis and precision medicine based on routine pathological images. The code for this project is publicly available on GitHub at `https://github.com/iclr2026stevs/stevs.v1.0`.

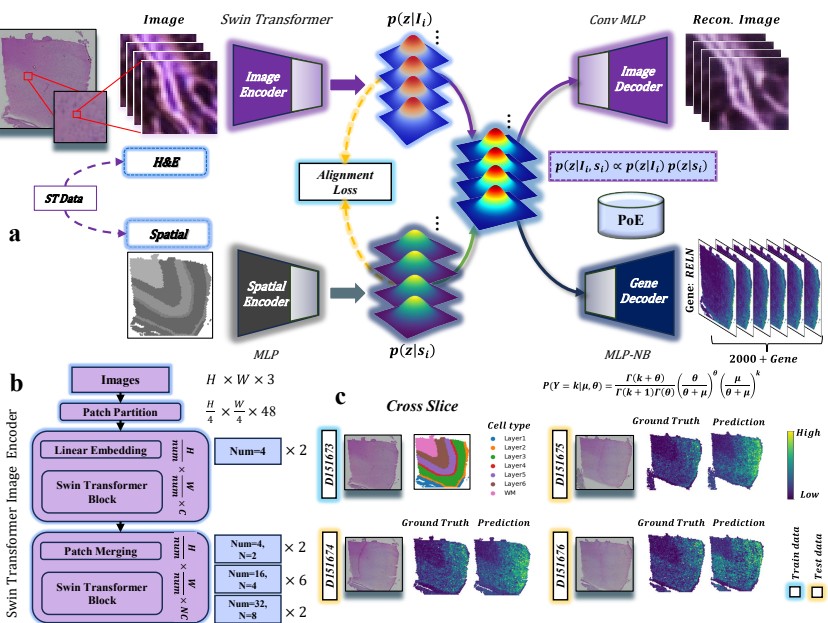

Figure 2: The STevs model architecture. (a) The overall framework of STevs. (b) The Swin Transformer architecture. (c) An example of the cross-slice prediction task.

## 2 RELATED WORK

Predicting spatially resolved transcriptomics (SRT) from histology images has become a significant area of research in computational pathology Long et al. (2023). Early pioneering works primarily employed CNNs, such as ST-Net He et al. (2020) and DeepSpaCE Monjo et al. (2022), to predict gene expression from individual image patches. While these methods successfully demonstrated the feasibility of the task, their patch-based, independent processing inherently ignored the crucial spatial context within the tissue Ståhl et al. (2016). To overcome this, subsequent research introduced GNNs to explicitly model the relationships between spots. Methods like SpaGCN Hu et al. (2021) and Hist2ST Zeng et al. (2022) construct spatial proximity graphs to aggregate neighborhood information. However, GNN-based approaches can be limited by their reliance on complex graph construction strategies and their difficulty in capturing long-range dependencies across the entire slide.

More recently, the field has shifted towards Vision Transformers (ViTs) and their variants, aimed at learning more powerful, long-range features directly from imagesHan et al. (2022). Initial applications like HisToGenePang et al. (2021) utilized a standard ViT architecture, while more advanced models such as iStarZhang et al. (2024) and iSCALE Schroeder et al. (2025) leverage hierarchical vision transformers (HVITs) to capture multi-scale visual features. Additionally, other learning paradigms have been explored, including generative models like STAGELi et al. (2024) and contrastive learning frameworks like BLEEPXie et al. (2023). Despite these significant advancements, most existing methods still struggle with two critical challenges: one is achieving robust generalization across unseen tissue slices from different batches or patients (Fig. 1d); the other is effectively fusing multimodal information (e.g., visual and spatial data) while properly accounting for the inherent uncertainty (Fig. 2a).

**Our Contributions**, STevs builds upon the aforementioned research and introduces innovations in several key aspects. First, unlike most discriminative models, we adopt a generative framework based on a multimodal VAE, which probabilistically fuses the two modalities of visual histology and spatial coordinates using a PoE, rather than simple feature concatenation Long et al. (2023), allowing the uncertainty weights of the two modalities to be implicitly determined by the data. Second, we leverage the powerful Swin Transformer to deeply mine the complex intra-spot visual context, thereby replacing the dependency on GNNsGao et al. (2024); Zeng et al. (2022); Yang et al. (2024) and manual hierarchical partitioning schemesChung et al. (2024); Wang et al. (2025). Finally, we are the first to introduce a latent space alignment loss, which explicitly encourages the model to learn a slide-invariant universal representation, enabling superior generalization ability on the highly challenging cross-slice prediction tasks.

## 3 METHODS

Unlike methods that rely on single information sources, simple feature concatenation Gao et al. (2024); He et al. (2020) or cross-attention mechanisms Xu et al. (2023), STevs accounts for the uncertainty of heterogeneous representations during modality fusion. The workflow uses a Multi-Modal Variational Autoencoder (MM-VAE) Suzuki et al. (2016); He et al. (2024) architecture with three core components (Fig. 2a): parallel encoders (**Swin Transformer** Liu et al. (2021) and **MLP** LeCun et al. (2015)), a **PoE** Hinton (2002) fusion mechanism, and multi-task decoders for image reconstruction and gene expression prediction using a **Negative Binomial (NB) distribution** Lopez et al. (2018).The detailed proof of our method's formulas is provided in Appendix A, the detailed architecture of the STevs model is described in Appendix B, and the step-by-step procedure for our method can be found in Appendix C.

### 3.1 PARALLEL MODALITY ENCODERS

To efficiently process the distinct data modalities, STevs employs two parallel encoders. **Image histology Encoder:** We choose a Swin Transformer as the image encoder, with its architecture detailed in Fig. 2b). The Swin Transformer was selected not only for its ability to capture long-range dependencies, characteristic of Transformer architectures, but also because its hierarchical design effectively extracts multi-scale visual features, which is crucial for identifying complex histopathological patterns. Compared to a standard ViT, its shifted window attention mechanism is more computationally efficient. Compared to CNNs, it better models global context, rather than being confined to limited receptive fields. We employ an ImageNet pre-trained Stickels et al. (2021) Swin Transformer Ji et al. (2020) that processes an input tensor of histology image patches ($I_i$) and outputs the parameters for the image latent distribution: a mean vector $\mu_{\text{img}}$ and a log variance vector $\log \sigma_{\text{img}}^2$. **Spatial Context Encoder:** For spatial coordinates, we utilize a concise MLP. While more complex spatial encoding schemes exist (e.g., using Fourier features Tancik et al. (2020)), we found that a simple MLP is sufficient to capture the absolute positional context Andersson et al. (2021) ($s_i = (x_i, y_i)$) for this task, while effectively avoiding overfitting to specific spatial patterns, thereby enhancing the model's generalization ability across different tissue slides. It similarly outputs parameters for the spatial latent distribution: $\mu_{\text{spatial}}$ and $\log \sigma_{\text{spatial}}^2$.

### 3.2 PROBABILISTIC FUSION AND LATENT SPACE ALIGNMENT

The latent distributions from the encoders are integrated through two key synergistic mechanisms. First, to fuse information from different modalities Gao et al. (2024), we moved beyond simple feature concatenation or cross-attention mechanisms Xu et al. (2023), as they cannot directly model the contributions and uncertainties of different information sources. Instead, we innovatively employ the PoE framework. The advantage of PoE lies in its ability to fuse the latent distributions of the image $\mathcal{N}(\mu_{\text{img}}, \sigma_{\text{img}}^2)$ and spatial $\mathcal{N}(\mu_{\text{spatial}}, \sigma_{\text{spatial}}^2)$ modalities at a probabilistic level Stickels et al. (2021). When both image and spatial information are clear, the fused posterior distribution becomes sharper (i.e., has smaller variance and higher certainty). Conversely, when one modality is ambiguous or noisy (e.g., a histological ly featureless tissue region), PoE automatically downweights its contribution to that prediction, yielding a more robust joint representation. This process yields a more precise joint posterior distribution $\mathcal{N}(\mu_{\text{fused}}, \sigma_{\text{fused}}^2)$, whose parameters are calculated analytically:

$$\sigma_{\text{fused}}^2 = \left( \frac{1}{\sigma_{\text{img}}^2} + \frac{1}{\sigma_{\text{spatial}}^2} \right)^{-1} , \quad \mu_{\text{fused}} = \left( \frac{\mu_{\text{img}}}{\sigma_{\text{img}}^2} + \frac{\mu_{\text{spatial}}}{\sigma_{\text{spatial}}^2} \right) \sigma_{\text{fused}}^2 \tag{1}$$

Concurrently, while PoE ensures effective fusion for an *individual data point*, learning a *slide-invariant* universal representation to address the domain shift problem in cross-slice prediction requires a global strategy. Inspired by prior work in multimodal representation learning Ji et al. (2020); Wagner et al. (2023), we introduce a **Latent Space Alignment Loss** ($\mathcal{L}_{\text{align}}$). This loss enhances the model's ability to learn universal representations by explicitly minimizing the Mean Squared Error (MSE) Kingma & Welling (2013) between the mean vectors of the two modalities. This forces the image and spatial encoders to learn a *semantically consistent shared latent space*, ensuring that similar spatial locations and cell types are mapped to nearby regions in the latent space regardless of histological variations. This is key to achieving strong generalization.

$$\mathcal{L}_{\text{align}} = \frac{1}{N} \sum_{i=1}^{N} \left\| z_{\text{img}}^{(i)} - z_{\text{spatial}}^{(i)} \right\|_2^2 \tag{2}$$

where $N$ is the spot number and $i$ is the index.

### 3.3 Multi-Task Decoders and Training Objective

From the fused posterior, a latent vector $z$ is sampled using the reparameterization trick Kingma & Welling (2013) and fed into our two decoders, which are designed as a multi-task learning framework. The overall model is trained by minimizing a composite loss function:

$$L_{\text{total}} = \lambda_{\text{img}} L_{\text{img}} + \lambda_{\text{rna}} L_{\text{rna}} + \beta L_{\text{KLD}} + \gamma L_{\text{align}} \tag{3}$$

The **Image Reconstruction Decoder** (composed of transposed convolutions) reconstructs the input image patch $\hat{I}_i$. This is a deliberate design choice: the image reconstruction task acts as a powerful regularizer, forcing the image encoder to learn information-rich visual features capable of preserving fine-grained tissue structures, rather than only abstract features sufficient for gene prediction. This enriched representation, in turn, improves the accuracy of the primary gene prediction task. Its loss, $L_{\text{img}}$, is the MSE between the original and reconstructed images Kingma & Welling (2013). The **Gene Expression Decoder** (an MLP) predicts the gene expression parameters. Considering the count-based nature and prevalent over-dispersion of spatial transcriptomics data Naik et al. (2020), we chose a NB distribution Lopez et al. (2018) to model the gene expression, which more accurately captures these statistical properties compared to MSE or a Poisson distribution, leading to more reliable predictions. Its loss, $L_{\text{rna}}$, is the Negative Log-Likelihood of the NB distribution. $L_{\text{KLD}}$ is the standard Kullback-Leibler (KL) divergence loss that regularizes the fused latent space to approximate a standard normal distribution Kingma & Welling (2013). We employ a KL annealing strategy Bowman et al. (2016) on its weight $\beta$ to prevent posterior collapse. The impact of the weights for each loss component on the model's performance is discussed in the Appendix I. In our experiments, we used default values of $\lambda_{\text{img}} = 1.0$, $\lambda_{\text{rna}} = 10.0$, $\beta = 0.5$, and $\gamma = 0.5$.

## 4 Experiments

### 4.1 Datasets

We evaluated our model on a total of 16 tissue sections from the public human dorsolateral prefrontal cortex (DLPFC) Maynard et al. (2021) and 10x Visium mouse brain Ståhl et al. (2016) datasets. For model training, we filtered for spatially variable genes (SVGs) using `scanpy` Wolf et al. (2018) and `squidpy` Palla et al. (2022), resulting in over 2,000 genes per group (Appendix E), and extracted corresponding image patches . To further assess generalization, we also used Human Breast Cancer (HBC) Wu et al. (2021) and Human Squamous Cell Carcinoma (HSC) Ji et al. (2020) datasets. Additionally, a MISAR-seq Jiang et al. (2023) dataset from different individuals at different time points was also used (Appendix J). All detailed data processing methods, patch extraction rules, and gene filtering criteria are provided in the Appendix D.

## 4.2 EXPERIMENTAL SETUP

We evaluated model performance under two settings: intra-slice and a more challenging cross-slice prediction. Intra-slice evaluation involved random data splitting within each slice, while cross-slice evaluation used a single-slice training scheme for cross-validation within each group (Fig. 2c). We quantified prediction accuracy using MSEKingma & Welling (2013), Pearson Correlation Coefficient (PCC)Pearson (1896), and Spearman's Rank Correlation Coefficient (SCC)Spearman (1987). We ran all experiments for 100 epochs with a learning rate of 1e-4 on four A100 (80GB) GPUs. All detailed training hyperparameters information are provided in the Appendix K.

Table 1: Intra-slice cross-validation performance of models across DLPFC and 10x Mouse Brain dataset groups. Metrics: MSE, PCC, SCC. Bold values indicate column-wise optimal performance (min MSE, max PCC/SCC). Standard deviations are omitted for space; full data in Appendix E. "Promotion" denotes the relative percentage improvement over the best-performing baseline model.

| Model Category | DLPFC Dataset | | | | | | | | | 10x Mouse Brain Dataset | | | | | |
| | Human 1 | | | Human 2 | | | Human 3 | | | Sagittal-Anterior | | | Sagittal-Posterior | | |
| | MSE ↓ | PCC ↑ | SCC ↑ | MSE ↓ | PCC ↑ | SCC ↑ | MSE ↓ | PCC ↑ | SCC ↑ | MSE ↓ | PCC ↑ | SCC ↑ | MSE ↓ | PCC ↑ | SCC ↑ |
|---|---|---|---|---|---|---|---|---|---|---|---|---|---|---|---|
| **Local Image-based** | | | | | | | | | | | | | | | |
| ST-Net (Nat. B.E. He et al. (2020)) | 1.494 | 0.033 | 0.070 | 1.348 | 0.053 | 0.086 | 0.365 | 0.123 | 0.134 | 0.896 | 0.051 | 0.066 | 0.915 | 0.043 | 0.121 |
| BLEEP (NeurIPS Xie et al. (2023)) | 1.551 | 0.036 | 0.037 | 1.365 | 0.058 | 0.057 | 1.067 | 0.086 | 0.077 | 0.772 | 0.086 | 0.087 | 0.967 | 0.123 | 0.117 |
| **Graph-based Context** | | | | | | | | | | | | | | | |
| EGN (PR Yang et al. (2024)) | 0.995 | 0.051 | 0.054 | 1.008 | 0.053 | 0.066 | 0.997 | 0.103 | 0.109 | 0.739 | 0.084 | 0.076 | 1.087 | 0.099 | 0.107 |
| IGI-DL (Cell R.M. Gao et al. (2024)) | 0.205 | 0.115 | 0.117 | 0.297 | 0.155 | 0.152 | 0.284 | 0.138 | 0.124 | 0.324 | 0.239 | 0.242 | 0.584 | 0.292 | 0.264 |
| **Transformer-based Context** | | | | | | | | | | | | | | | |
| iStar (Nat. Biot. Zhang et al. (2024)) | 0.149 | 0.191 | 0.188 | 0.194 | 0.204 | 0.229 | 0.171 | 0.236 | 0.230 | 0.254 | 0.384 | 0.375 | 0.264 | 0.459 | 0.397 |
| TRIPLEX (CVPR Chung et al. (2024)) | 0.181 | 0.131 | 0.125 | 0.211 | 0.194 | 0.186 | 0.179 | 0.211 | 0.199 | 0.372 | 0.232 | 0.216 | 0.345 | 0.315 | 0.297 |
| M2ORT (AAAI Wang et al. (2025)) | 1.000 | -0.001 | -0.000 | 1.006 | -0.001 | -0.000 | 1.019 | -0.000 | -0.000 | 1.008 | 0.001 | 0.001 | 1.020 | 0.001 | 0.001 |
| **Coordinate-based Generative** | | | | | | | | | | | | | | | |
| STAGE (NAR Li et al. (2024)) | 0.259 | 0.108 | 0.105 | 0.307 | 0.139 | 0.130 | 0.339 | 0.150 | 0.149 | 0.462 | 0.104 | 0.094 | 0.502 | 0.120 | 0.123 |
| **STevs (Ours)** | **0.142** | **0.215** | **0.202** | **0.188** | **0.281** | **0.271** | **0.166** | **0.296** | **0.263** | **0.239** | **0.413** | **0.396** | **0.208** | **0.486** | **0.423** |
| **Promotion** | 4.7% | 12.6% | 7.4% | 3.1% | 37.7% | 18.3% | 2.9% | 25.4% | 14.3% | 5.9% | 7.6% | 5.6% | 21.2% | 5.9% | 6.5% |

## 4.3 MAIN PERFORMANCE

In the intra-slice cross-validation setting, as shown in Table 1, STevs demonstrates highly competitive performance, achieving the lowest MSE and the highest PCC and SCC across all five dataset groups. This indicates that STevs is a top-performing model in standard single-sample learning tasks. However, intra-slice testing cannot effectively evaluate a model's generalization ability when faced with unseen slices from new patients, batches, or different experimental conditions. For instance, iStar, one of the strongest baselines in the intra-slice setting, exhibits a steep performance decline when transitioning to the cross-slice task. Its PCC drops from 0.204 to 0.105, and its SCC drops from 0.224 to 0.109 (data from Table 1 and Table 2, respectively), a performance decay of nearly 50%.In stark contrast, STevs displays excellent and robust performance in the demanding cross-slice setting. As shown in Table 2, STevs significantly surpasses all baseline models across all metrics on all datasets. Its superiority is particularly prominent on the Human 3 dataset, where STevs achieves improvements of 109.8% in PCC and 95.8% in SCC over the next-best model. These results provide strong evidence that STevs successfully learns a transferable, slice-invariant histology-to-gene mapping, equipping it with the generalization capability required for real-world applications. Further details on this section are provided in Appendix E. We also demonstrated the superiority of our model in extended experiments on the HBC and HSC datasets, with further details available in Appendix J.

## 4.4 ABLATION STUDIES

We conducted comprehensive ablation studies to validate our key design choices, with results summarized in Table 3 (intra-slice) and Table 4 (cross-slice). The results underscore the necessity of each core component: removing the spatial encoder (STevs w/o Spatial Encoder) or the latent space alignment loss (STevs w/o Alignment Loss) critically impairs cross-slice generalization, while image reconstruction (STevs w/o Image Decoder) acts as an effective regularizer. Our proposed PoE fusion mechanism demonstrated superior performance over common alternatives including feature concatenation Baltrušaitis et al. (2018), deterministic fusion, and cross-attention Xu et al. (2023). Architectural evaluations confirmed the Swin Transformer's superiority over ViT Han et al. (2022) and CNN Krizhevsky et al. (2012) backbones, and the robustness of our simple MLP spatial encoder compared to more complex Gaussian Process (GP) Williams & Rasmussen

Table 2: Comparison of cross-slice cross-validation model performance across dataset groups of DLPFC and 10x Mouse Brain. Metrics: MSE, PCC, SCC. Full data available in the Appendix E.

| Model Category | DLPFC Dataset | | | | | | | | | 10x Mouse Brain Dataset | | | | | |
| | Human 1 | | | Human 2 | | | Human 3 | | | Sagittal-Anterior | | | Sagittal-Posterior | | |
| | MSE↓ | PCC↑ | SCC↑ | MSE↓ | PCC↑ | SCC↑ | MSE↓ | PCC↑ | SCC↑ | MSE↓ | PCC↑ | SCC↑ | MSE↓ | PCC↑ | SCC↑ |
|---|---|---|---|---|---|---|---|---|---|---|---|---|---|---|---|
| **Local Image-based** | | | | | | | | | | | | | | | |
| ST-Net (Nat. B.E. He et al. (2020)) | 1.471 | 0.009 | 0.062 | 1.571 | 0.008 | 0.063 | 1.283 | 0.040 | 0.043 | 1.861 | 0.010 | 0.052 | 1.502 | 0.071 | 0.131 |
| BLEEP (NeurIPS Xie et al. (2023)) | 1.758 | 0.029 | 0.030 | 1.274 | 0.039 | 0.036 | 1.574 | 0.039 | 0.034 | 1.436 | 0.069 | 0.067 | 1.229 | 0.118 | 0.111 |
| **Graph-based Context** | | | | | | | | | | | | | | | |
| EGN (PR Yang et al. (2024)) | 0.905 | 0.049 | 0.056 | 0.896 | 0.052 | 0.045 | 0.937 | 0.052 | 0.052 | 1.159 | 0.084 | 0.075 | 0.825 | 0.117 | 0.130 |
| IGI-DL (Cell R.M. Gao et al. (2024)) | 0.717 | 0.059 | 0.059 | 1.859 | 0.029 | 0.030 | 1.908 | 0.008 | 0.001 | 0.918 | 0.089 | 0.087 | 0.924 | 0.118 | 0.126 |
| **Transformer-based Context** | | | | | | | | | | | | | | | |
| iStar (Nat. Biot. Zhang et al. (2024)) | 0.262 | 0.126 | 0.136 | 0.215 | 0.105 | 0.109 | 0.319 | 0.122 | 0.118 | 0.273 | 0.301 | 0.300 | 0.269 | 0.363 | 0.325 |
| TRIPLEX (CVPR Chung et al. (2024)) | 0.487 | 0.097 | 0.092 | 0.566 | 0.083 | 0.083 | 0.814 | 0.071 | 0.069 | 0.438 | 0.197 | 0.180 | 0.450 | 0.256 | 0.247 |
| M2ORT (AAAI Wang et al. (2025)) | 1.205 | 0.005 | 0.005 | 1.188 | -0.004 | -0.002 | 1.106 | -0.001 | 0.001 | 1.133 | 0.006 | 0.007 | 1.253 | 0.001 | 0.001 |
| **Coordinate-based Generative** | | | | | | | | | | | | | | | |
| STAGE (NAR Li et al. (2024)) | 1.186 | 0.044 | 0.042 | 0.921 | 0.046 | 0.047 | 0.615 | 0.074 | 0.077 | 0.624 | 0.125 | 0.118 | 0.631 | 0.156 | 0.158 |
| **STevs (Ours)** | **0.145** | **0.153** | **0.152** | **0.202** | **0.167** | **0.166** | **0.174** | **0.256** | **0.231** | **0.261** | **0.362** | **0.350** | **0.223** | **0.442** | **0.392** |
| **Promotion** | 44.7% | 21.4% | 11.8% | 6.0% | 59.0% | 52.3% | 45.5% | 109.8% | 95.8% | 4.4% | 20.3% | 16.7% | 17.1% | 21.8% | 20.6% |

(2006) or Fourier Feature-based Mildenhall et al. (2021) encoders. Finally, leveraging pre-trained weights (`STevs w/o Pretrained`) consistently improved performance. These findings collectively validate the design of STevs.

To systematically validate the necessity of each core component within the STevs model and to demonstrate the superiority of our design choices, we conducted a series of comprehensive ablation studies. We evaluated the impact on performance by removing or replacing the model's key modules, with the results summarized in Table 3 (intra-slice) and Table 4 (cross-slice). Further details can be found in Appendix F.

Table 3: Intra-slice cross-validation Performance Comparison of STevs Variants on DLPFC (Human) and 10x Mouse Brain (Sagittal) Datasets

| Model Variant | Human 1 | | | Human 2 | | | Human 3 | | | Sagittal-Anterior | | | Sagittal-Posterior | | |
| | MSE↓ | PCC↑ | SCC↑ | MSE↓ | PCC↑ | SCC↑ | MSE↓ | PCC↑ | SCC↑ | MSE↓ | PCC↑ | SCC↑ | MSE↓ | PCC↑ | SCC↑ |
|---|---|---|---|---|---|---|---|---|---|---|---|---|---|---|---|
| *Component Ablation* | | | | | | | | | | | | | | | |
| STevs w/o Image Decoder | 0.145 | 0.211 | 0.199 | 0.190 | 0.278 | 0.268 | 0.169 | 0.291 | 0.259 | 0.242 | 0.409 | 0.392 | 0.210 | 0.481 | 0.419 |
| STevs w/o Spatial Encoder | 0.171 | 0.172 | 0.162 | 0.226 | 0.225 | 0.217 | 0.199 | 0.237 | 0.210 | 0.287 | 0.330 | 0.317 | 0.250 | 0.389 | 0.338 |
| STevs w/o Alignment Loss | 0.147 | 0.209 | 0.200 | 0.188 | 0.280 | 0.266 | 0.172 | 0.289 | 0.261 | 0.241 | 0.411 | 0.390 | 0.213 | 0.479 | 0.421 |
| *Fusion Mechanism Ablation* | | | | | | | | | | | | | | | |
| STevs (Concat) | 0.146 | 0.209 | 0.197 | 0.191 | 0.276 | 0.265 | 0.170 | 0.288 | 0.257 | 0.244 | 0.407 | 0.388 | 0.212 | 0.478 | 0.415 |
| STevs (Deterministic) | **0.141** | 0.212 | 0.200 | 0.193 | 0.272 | 0.261 | 0.173 | 0.285 | 0.253 | 0.247 | 0.401 | 0.384 | 0.214 | 0.472 | 0.410 |
| STevs (Cross-Attention) | 0.143 | 0.213 | 0.201 | **0.187** | **0.283** | **0.273** | 0.167 | 0.294 | 0.261 | 0.240 | 0.411 | 0.394 | **0.206** | 0.488 | **0.425** |
| *Spatial Encoder Variants* | | | | | | | | | | | | | | | |
| STevs (Gaussian Process) | 0.144 | 0.212 | 0.200 | 0.189 | 0.279 | 0.269 | 0.168 | 0.293 | 0.260 | 0.240 | 0.410 | 0.393 | 0.209 | 0.483 | 0.420 |
| STevs (MLP w/ Fourier) | 0.145 | 0.210 | 0.198 | 0.191 | 0.276 | 0.265 | 0.170 | 0.290 | 0.258 | **0.238** | **0.415** | **0.399** | 0.211 | 0.480 | 0.417 |
| *Architecture Variants* | | | | | | | | | | | | | | | |
| STevs (Convolutional) | 0.217 | 0.163 | 0.162 | 0.259 | 0.215 | 0.203 | 0.246 | 0.224 | 0.203 | 0.351 | 0.322 | 0.307 | 0.332 | 0.382 | 0.313 |
| STevs (ViT) | 0.149 | 0.210 | 0.194 | 0.199 | 0.271 | 0.266 | 0.176 | 0.290 | 0.252 | 0.251 | 0.403 | 0.391 | 0.217 | 0.476 | 0.414 |
| STevs w/o Pretrained | 0.191 | 0.176 | 0.173 | 0.243 | 0.231 | 0.218 | 0.223 | 0.239 | 0.213 | 0.318 | 0.347 | 0.329 | 0.290 | 0.403 | 0.357 |
| **STevs (Ours)** | 0.142 | **0.215** | **0.202** | 0.188 | 0.281 | 0.271 | **0.166** | **0.296** | **0.263** | 0.239 | 0.413 | 0.396 | 0.208 | **0.489** | 0.423 |

Table 4: Cross-slice cross-validation Performance Comparison of STevs Variants on DLPFC (Human) and 10x Mouse Brain (Sagittal) Datasets

| Model Variant | Human 1 | | | Human 2 | | | Human 3 | | | Sagittal-Anterior | | | Sagittal-Posterior | | |
| | MSE↓ | PCC↑ | SCC↑ | MSE↓ | PCC↑ | SCC↑ | MSE↓ | PCC↑ | SCC↑ | MSE↓ | PCC↑ | SCC↑ | MSE↓ | PCC↑ | SCC↑ |
|---|---|---|---|---|---|---|---|---|---|---|---|---|---|---|---|
| *Component Ablation* | | | | | | | | | | | | | | | |
| STevs w/o Image Decoder | 0.175 | 0.131 | 0.128 | 0.234 | 0.145 | 0.142 | 0.245 | 0.132 | 0.130 | 0.298 | 0.320 | 0.305 | 0.263 | 0.401 | 0.350 |
| STevs w/o Spatial Encoder | 0.345 | 0.115 | 0.120 | 0.360 | 0.110 | 0.112 | 0.380 | 0.101 | 0.105 | 0.433 | 0.212 | 0.209 | 0.351 | 0.280 | 0.263 |
| STevs w/o Alignment Loss | 0.158 | 0.145 | 0.142 | 0.225 | 0.150 | 0.148 | 0.238 | 0.141 | 0.138 | 0.285 | 0.331 | 0.315 | 0.249 | 0.408 | 0.360 |
| *Fusion Mechanism Ablation* | | | | | | | | | | | | | | | |
| STevs (Concat) | 0.225 | 0.115 | 0.112 | 0.240 | 0.110 | 0.108 | 0.264 | 0.102 | 0.100 | 0.381 | 0.270 | 0.258 | 0.325 | 0.349 | 0.311 |
| STevs (Deterministic) | 0.241 | 0.109 | 0.106 | 0.255 | 0.104 | 0.101 | 0.282 | 0.098 | 0.095 | 0.399 | 0.255 | 0.243 | 0.350 | 0.328 | 0.302 |
| STevs (Cross-Attention) | 0.155 | 0.148 | 0.145 | 0.242 | 0.134 | 0.133 | 0.255 | 0.128 | 0.120 | 0.313 | 0.290 | 0.280 | 0.268 | 0.354 | 0.314 |
| *Spatial Encoder Variants* | | | | | | | | | | | | | | | |
| STevs (Gaussian Process) | 0.335 | 0.125 | 0.128 | 0.355 | 0.118 | 0.122 | 0.375 | 0.110 | 0.115 | 0.425 | 0.218 | 0.214 | 0.345 | 0.287 | 0.270 |
| STevs (MLP w/ Fourier) | 0.330 | 0.128 | 0.130 | 0.351 | 0.121 | 0.125 | 0.370 | 0.113 | 0.118 | 0.421 | 0.223 | 0.219 | 0.340 | 0.291 | 0.275 |
| *Architecture Variants* | | | | | | | | | | | | | | | |
| STevs (Convolutional) | 0.265 | 0.141 | 0.138 | 0.280 | 0.135 | 0.131 | 0.295 | 0.125 | 0.120 | 0.398 | 0.275 | 0.264 | 0.350 | 0.315 | 0.298 |
| STevs (ViT) | 0.166 | 0.140 | 0.134 | 0.232 | 0.145 | 0.147 | 0.241 | 0.139 | 0.135 | 0.318 | 0.301 | 0.309 | 0.261 | 0.394 | 0.356 |
| STevs w/o Pretrained | 0.254 | 0.099 | 0.101 | 0.278 | 0.091 | 0.095 | 0.300 | 0.085 | 0.088 | 0.413 | 0.237 | 0.226 | 0.370 | 0.319 | 0.286 |
| **STevs (Ours)** | **0.145** | **0.153** | **0.152** | **0.202** | **0.167** | **0.166** | **0.174** | **0.256** | **0.231** | **0.261** | **0.362** | **0.350** | **0.223** | **0.442** | **0.392** |

### 4.5 IN-DEPTH ANALYSIS OF MODEL GENERALIZATION AND ROBUSTNESS

#### 4.5.1 QUALITATIVE ANALYSIS OF GENE EXPRESSION PREDICTION

To visually evaluate the model's generalization ability, we visualized the predicted expression for the key gene OLFM1 Maynard et al. (2021); Shen et al. (2025) on the Human3 dataset group. In the stringent cross-slice prediction task (Figure 3), nearly all baseline models fail completely. Their predictions are indistinguishable from noise, often yielding negative SCC. In stark contrast, STevs is the only method that accurately reconstructs the complex layered structure of OLFM1 on unseen slices while maintaining a high spatial correlation (SCC > 0.56), demonstrating its superior generalization performance. Results from other dataset groups are available in the Appendix G.

#### 4.5.2 LATENT SPACE VISUALIZATION REVEALS EFFECTIVE DOMAIN ADAPTATION

To investigate the source of the model's generalization ability, we visualized the latent space learned from unseen slices using UMAP (Figure 4). The results show that the Fused Latent space successfully eliminates inter-slice batch effects (top row) while accurately preserving the true biological structure (bottom row). In contrast, the Image Latent space exhibits noticeable batch effects, and the Spatial Latent space fails to effectively distinguish the biological structures. This demonstrates that STevs learns a slide-invariant, universal representation through its PoE fusion, which is a key factor in the model's generalization ability.

#### 4.5.3 MODEL ROBUSTNESS UNDER SINGLE-MODALITY INFERENCE

To validate the model's robustness with incomplete information, we evaluated its single-modality inference performance (Table 5). The results demonstrate that the model remains robust even under adverse conditions with only image or coordinate inputs. Notably, in the cross-slice task, the performance of the single-modality STevs still surpasses that of most fully-equipped baseline models, which strongly demonstrates the model's exceptional robustness. Furthermore, the robust performance using only spatial coordinates suggests that our model can also be applied to super-resolution tasks.

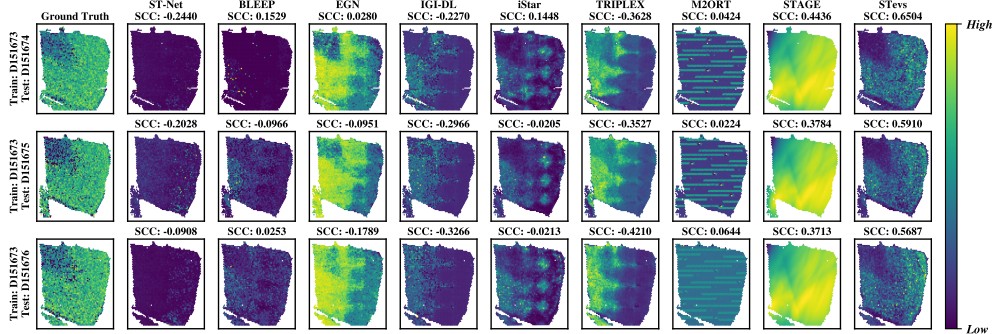

Figure 3: The cross-slice validation results of the OLFM1 gene on the other 3 slices of human3, with D151673 used as the training set

Table 5: Performance Evaluation of STevs Using Single Modality for Inference

| Inference Mode | Human 1 | | | Human 2 | | | Human 3 | | | Sagittal-Anterior | | | Sagittal-Posterior | | |
|---|---|---|---|---|---|---|---|---|---|---|---|---|---|---|---|
| | MSE ↓ | PCC ↑ | SCC ↑ | MSE ↓ | PCC ↑ | SCC ↑ | MSE ↓ | PCC ↑ | SCC ↑ | MSE ↓ | PCC ↑ | SCC ↑ | MSE ↓ | PCC ↑ | SCC ↑ |
| Image-only (Intra-slice) | 0.155 | 0.201 | 0.190 | 0.203 | 0.265 | 0.254 | 0.181 | 0.279 | 0.248 | 0.260 | 0.391 | 0.375 | 0.224 | 0.463 | 0.405 |
| Image-only (Cross-slice) | 0.189 | 0.130 | 0.128 | 0.258 | 0.141 | 0.139 | 0.223 | 0.215 | 0.198 | 0.334 | 0.302 | 0.291 | 0.287 | 0.388 | 0.344 |
| Spatial-only | 0.301 | 0.115 | 0.111 | 0.325 | 0.123 | 0.119 | 0.312 | 0.188 | 0.170 | 0.391 | 0.285 | 0.258 | 0.346 | 0.301 | 0.319 |

### 4.6 ACCURATE RECOVERY OF SPATIAL DOMAINS

As shown in Figure 5, benchmarked against manual annotations , the clustering Adjusted Rand Index (ARI)Hubert & Arabie (1985) score from STevs's predictions (0.2098) not only surpasses

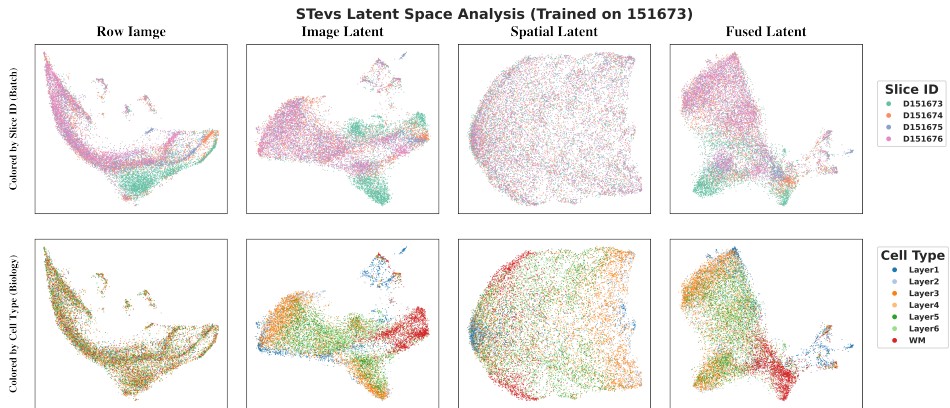

Figure 4: Latent Space Visualization Reveals Effective Domain Adaptation

the baseline iStar (0.0995) but even exceeds the clustering result from the ground truth expression profile itself (0.1692). This suggests that STevs's predictions not only faithfully reconstruct the unseen expression profiles from the image but may also serve a denoising function, thereby enabling a more accurate recovery of the tissue architecture. Further details can be found in Appendix H.

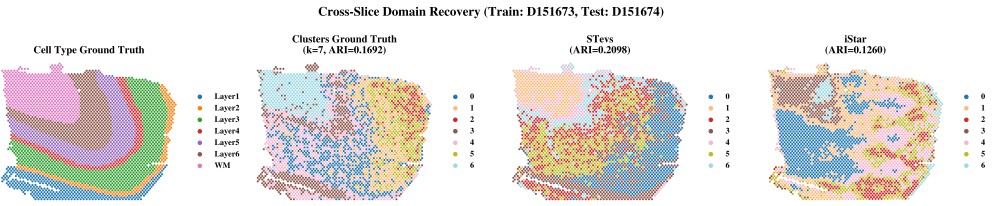

Figure 5: Comparison of Cross-Slice Spatial Domain Recovery Based on Predicted Expression Profiles

## 5 CONCLUSION

This paper introduces STevs, a deep generative framework designed to address the generalization challenge in cross-slice gene expression prediction for SRT data. The model employs a Product of Experts mechanism to probabilistically fuse visual and spatial representations, while simultaneously learning a unified, slice-invariant representation by leveraging a latent space alignment loss.. Extensive experiments demonstrate that STevs not only outperforms in intra-slice tasks but also exhibits significantly superior cross-slice generalization ability compared to existing methods. Our work provides a powerful tool for large-scale, low-cost virtual spatial transcriptomics analysis, showcasing its immense potential for biomedical research and future clinical applications.

## 6 DISCUSSION

The key to STevs' success lies in learning a representation that is robust to slice-level uncertainty. By leveraging probabilistic fusion and an alignment loss, it effectively overcomes inter-slice information discrepancies associated with spatial context, capturing the essential relationship between histology and gene expression, which is crucial for processing clinical samples from diverse sources. Despite its strong performance, STevs has certain limitations. In particular, it is more suitable for 3D serial sections or slices originating from the same organ, as a certain degree of spatial similarity across slices is required. In future work, we plan to use spatio-temporal Gaussian process modeling Williams & Rasmussen (2006) to enhance the model's generalization capability.

REPRODUCIBILITY STATEMENT

To ensure the reproducibility of this research, we provide the complete code, experimental setup, and data processing steps. Our implementation, developed using the PyTorch framework, has been released via an anonymous GitHub link. The datasets used in this work are all publicly available; we provide detailed descriptions and data preprocessing in Appendix D. Hyperparameter settings and details of the computational environment (including hardware specifications) can be found in Appendix B and D.2.3 to ensure that the experiments can be precisely reproduced. For the theoretical parts of the paper, we provide complete proofs and derivations in Appendix A and C.

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
