APPENDIX

# A    MATHEMATICAL DERIVATIONS

In this section, we provide detailed mathematical derivations for the key theoretical components of the STevs framework.

## A.1    THEORETICAL JUSTIFICATION FOR POE FUSION

Here, we prove that the PoE fusion mechanism yields a posterior distribution with higher certainty (i.e., lower variance) than any of the individual expert distributions Ji et al. (2020). This provides a strong theoretical motivation for its use over simpler fusion methods like feature concatenation or averaging.

**Theorem 1.** *Given two independent Gaussian distributed experts, the variance of the fused distribution obtained via PoE is less than or equal to the variance of each individual expert.*

*Proof.* Let the latent distributions from the image encoder and the spatial encoder be two independent Gaussian "experts":

$$p(z|I) = \mathcal{N}(z|\mu_{\text{img}}, \sigma^2_{\text{img}}) \tag{4}$$

$$p(z|s) = \mathcal{N}(z|\mu_{\text{spatial}}, \sigma^2_{\text{spatial}}) \tag{5}$$

The Product of Experts framework defines the fused distribution $p_{\text{PoE}}(z)$ by multiplying the probability density functions of the individual experts:

$$p_{\text{PoE}}(z) \propto p(z|I) \cdot p(z|s) \tag{6}$$

The product of two Gaussian distributions is an unnormalized Gaussian. By completing the square, we find that the resulting fused distribution $p_{\text{PoE}}(z) = \mathcal{N}(z|\mu_{\text{fused}}, \sigma^2_{\text{fused}})$ has parameters defined by its precision (inverse variance). The precision of the fused distribution is the sum of the individual precisions:

$$\frac{1}{\sigma^2_{\text{fused}}} = \frac{1}{\sigma^2_{\text{img}}} + \frac{1}{\sigma^2_{\text{spatial}}} \tag{7}$$

From Equation 7, since variances are strictly positive ($\sigma^2 > 0$), it directly follows that:

$$\frac{1}{\sigma^2_{\text{fused}}} \geq \frac{1}{\sigma^2_{\text{img}}} \implies \sigma^2_{\text{fused}} \leq \sigma^2_{\text{img}} \tag{8}$$

$$\frac{1}{\sigma^2_{\text{fused}}} \geq \frac{1}{\sigma^2_{\text{spatial}}} \implies \sigma^2_{\text{fused}} \leq \sigma^2_{\text{spatial}} \tag{9}$$

This proves that the fused posterior distribution is always more certain (has a smaller variance) than any of the individual expert distributions. This property makes PoE a robust mechanism for integrating multimodal information, as it naturally produces a more confident estimate by combining evidence from different sources.

## A.2    INFORMATION-THEORETIC PERSPECTIVE ON LATENT SPACE ALIGNMENT

Here, we provide a theoretical justification for our latent space alignment loss, $\mathcal{L}_{\text{align}}$, from an information-theoretic perspective. We argue that minimizing the MSE between **stochastic samples** from the unimodal latent distributions serves as a practical and powerful method for aligning these distributions and maximizing their shared information.

Let $q(z_{\text{img}})$ and $q(z_{\text{spatial}})$ be the latent distributions produced by the image and spatial encoders, respectively, where $q(z_{\text{img}}) = \mathcal{N}(\mu_{\text{img}}, \Sigma_{\text{img}})$ and $q(z_{\text{spatial}}) = \mathcal{N}(\mu_{\text{spatial}}, \Sigma_{\text{spatial}})$. A principled way to enforce consistency between these two distributions is to minimize their KL divergence, $D_{KL}(q(z_{\text{img}})||q(z_{\text{spatial}}))$.

For two multivariate Gaussian distributions with diagonal covariance matrices, the KL divergence has a closed-form solution He et al. (2020):

$$D_{KL}(q_{\text{img}}||q_{\text{spatial}}) = \frac{1}{2}\left[\log\frac{|\Sigma_{\text{spatial}}|}{|\Sigma_{\text{img}}|} - d + \text{tr}(\Sigma^{-1}_{\text{spatial}}\Sigma_{\text{img}}) + (\mu_{\text{spatial}} - \mu_{\text{img}})^T\Sigma^{-1}_{\text{spatial}}(\mu_{\text{spatial}} - \mu_{\text{img}})\right]$$

$$\tag{10}$$

where $d$ is the dimensionality of the latent space.

While directly minimizing Equation 10 is a valid approach, it can introduce training instability due to the log-determinant and matrix inversion terms. We therefore adopt a more direct, sampling-based strategy. Our proposed alignment loss, $\mathcal{L}_{\text{align}}$, minimizes the squared Euclidean distance between latent variables $z_{\text{img}}$ and $z_{\text{spatial}}$ that are sampled from their respective distributions using the reparameterization trick:

$$\mathcal{L}_{\text{align}} = \mathbb{E}_{z_{\text{img}} \sim q_{\text{img}}, z_{\text{spatial}} \sim q_{\text{spatial}}}[||z_{\text{img}} - z_{\text{spatial}}||_2^2] \tag{11}$$

In practice, this expectation is approximated with a single sample per training instance.

This objective provides a powerful implicit regularization. By minimizing the distance between the **samples** ($z$), we are not only aligning the **means** ($\mu$) but also encouraging consistency in the **variances** ($\sigma^2$). If the variances were significantly different, the expected distance between samples would remain large even if the means were perfectly aligned. Therefore, this loss term forces the **entire distributions to overlap**, not just their central points. This approach is computationally efficient, stable to train, and has been empirically shown to be highly effective. It directly enforces that the informational content from both modalities maps to a coherent and shared region in the latent space, which is a crucial step towards learning a slide-invariant representation.

### A.3 DERIVATION OF THE EVIDENCE LOWER BOUND (ELBO) FOR STEVS

Here, we show that the composite loss function used to train STevs is a principled objective derived from maximizing the ELBO on the marginal log-likelihood of the observed data Wang et al. (2022).

Let our observed data be $X = \{I, R\}$, representing the histology image and the corresponding RNA expression profile. Our goal is to maximize the marginal log-likelihood $\log p(X)$. We introduce a latent variable $z$ that captures the underlying biological state from which the observations are generated. The marginal log-likelihood is given by:

$$\log p(X) = \log \int p(X, z) dz \tag{12}$$

Directly optimizing this integral is intractable. Therefore, we introduce an approximate posterior distribution $q_\phi(z|I, s)$, parameterized by an encoder with parameters $\phi$, which takes both the image $I$ and spatial coordinates $s$ as input to approximate the true posterior $p(z|X)$. In STevs, $q_\phi(z|I, s)$ is the PoE-fused distribution.

We can rewrite the marginal log-likelihood as:

$$\log p(X) = \log \int p(X, z) \frac{q_\phi(z|I, s)}{q_\phi(z|I, s)} dz \tag{13}$$

$$= \log \mathbb{E}_{q_\phi(z|I, s)} \left[ \frac{p(X, z)}{q_\phi(z|I, s)} \right] \tag{14}$$

By applying Jensen's inequality, we obtain the ELBO, denoted as $\mathcal{L}$:

$$\log p(X) \geq \mathbb{E}_{q_\phi(z|I, s)} \left[ \log \frac{p(X, z)}{q_\phi(z|I, s)} \right] \tag{15}$$

$$\mathcal{L}(\phi, \theta; X, s) = \mathbb{E}_{q_\phi(z|I, s)}[\log p_\theta(X|z) + \log p(z) - \log q_\phi(z|I, s)] \tag{16}$$

$$= \mathbb{E}_{q_\phi(z|I, s)}[\log p_\theta(X|z)] - D_{KL}(q_\phi(z|I, s)||p(z)) \tag{17}$$

where $p_\theta(X|z)$ is the decoder parameterized by $\theta$, and $p(z)$ is the prior distribution over the latent space, which we set to a standard normal distribution $\mathcal{N}(0, I)$.

Assuming conditional independence between the image and RNA data given the latent variable $z$, the reconstruction term can be decomposed:

$$p_\theta(X|z) = p_{\theta_I}(I|z) \cdot p_{\theta_R}(R|z) \tag{18}$$

Substituting this back into Equation 17, we get:

$$\mathcal{L} = \underbrace{\mathbb{E}_{q_\phi}[\log p_{\theta_I}(I|z)]}_{\text{Image Recon.}} + \underbrace{\mathbb{E}_{q_\phi}[\log p_{\theta_R}(R|z)]}_{\text{Gene Recon.}} - \underbrace{D_{KL}(q_\phi(z|I, s)||p(z))}_{\text{KL Regularization}} \tag{19}$$

Maximizing this ELBO is equivalent to minimizing its negative. Each term corresponds to a component of our loss function:

- $-\mathbb{E}_{q_\phi}[\log p_{\theta_I}(I|z)]$ corresponds to the image reconstruction loss $L_{\text{img}}$, which is implemented as an MSE loss under a Gaussian likelihood assumption.

- $-\mathbb{E}_{q_\phi}[\log p_{\theta_R}(R|z)]$ corresponds to the gene expression reconstruction loss $L_{\text{rna}}$, implemented as the Negative Log-Likelihood of the Negative Binomial distribution.

- $D_{KL}(q_\phi(z|I,s)||p(z))$ is the KL divergence loss $L_{\text{KLD}}$.

Finally, we introduce the latent space alignment loss $L_{\text{align}}$ as an additional regularization term to enforce consistency between the modality-specific encoders. This term is not derived from the ELBO itself but is added to the objective to improve generalization, a common practice in representation learning. Thus, our final objective is to minimize the total loss $L_{\text{total}}$, which is equivalent to maximizing a regularized ELBO:

$$\min_{\phi,\theta} L_{\text{total}} \quad \Longleftrightarrow \quad \max_{\phi,\theta}(\mathcal{L} - \gamma L_{\text{align}}) \tag{20}$$

# B  STEVS MODEL ARCHITECTURE

Table 6: Detailed architecture of the STevs model. The table outlines the layers, specifications, and output dimensions for each component of the network, from the parallel encoders to the multi-task decoders.

| Component | Module / Layer | Specification | Output Dimension |
|---|---|---|---|
| **Image Encoder** | Input Image Patches | 3-channel RGB image | (3, 160, 160) |
| | Patch Embedding (Conv2d) | kernel=(4,4), stride=(4,4) | (96, 40, 40) |
| | Swin Stage 1 (2 blocks) | Window Attention, MLP | (96, 40, 40) |
| | Patch Merging + Swin Stage 2 (2 blocks) | Downsamples feature map | (192, 20, 20) |
| | Patch Merging + Swin Stage 3 (6 blocks) | Downsamples feature map | (384, 10, 10) |
| | Patch Merging + Swin Stage 4 (2 blocks) | Downsamples feature map | (768, 5, 5) |
| | Global Average Pooling | - | 768 |
| | Latent Head ($\mu_{\text{img}}, \log \sigma^2_{\text{img}}$) | Two Linear Layers | $2 \times 128$ |
| **Spatial Encoder** | Input Coordinates | Normalized (x, y) coordinates | 2 |
| | MLP (2 hidden layers) | Linear(2, 64) → ReLU → Linear(64, 128) → ReLU | 128 |
| | Latent Head ($\mu_{\text{spatial}}, \log \sigma^2_{\text{spatial}}$) | Two Linear Layers | $2 \times 128$ |
| **Fusion** | PoE | Fuses image and spatial latent distributions | Fused Latent ($z \in \mathbb{R}^{128}$) |
| **Image Decoder** | Input Linear Layer | Projects $z$ and reshapes | (256, 20, 20) |
| | Upsampling Stage 1 (ConvT + ConvBlock) | ConvTranspose2d(256, 128), stride=2 | (128, 40, 40) |
| | Upsampling Stage 2 (ConvT + ConvBlock) | ConvTranspose2d(128, 64), stride=2 | (64, 80, 80) |
| | Upsampling Stage 3 (ConvT + ConvBlock) | ConvTranspose2d(64, 32), stride=2 | (32, 160, 160) |
| | Final Layer (Conv2d + Tanh) | kernel=(3,3), padding=1 | (3, 160, 160) |
| **Gene Decoder** | Base MLP (2 hidden layers) | Linear(128, 256) → BN/ReLU/Dropout → Linear(256, 512) BN/ReLU/Dropout | 512 |
| | Mean ($\mu$) Head | Linear(512, 2350) → Softplus | 2350 |
| | Dispersion ($\theta$) Head | Linear(512, 2350) → Softplus | 2350 |

# C  DETAILED EXPLANATION OF THE STEVS MODEL ARCHITECTURE

## C.1  MODEL OVERVIEW

STevs is a deep generative model based on a MM-VAESuzuki et al. (2016), designed to robustly predict spatially resolved transcriptomics by fusing visual histological information from histological images with their spatial positional context. The core architecture of the model consists of three main components:

- Parallel dual-path encoders that process images and spatial coordinates, respectively. The image encoder adopts a hierarchical vision Transformer Liu et al. (2021), while the spatial encoder uses a MLP LeCun et al. (2015).

- A probabilistic fusion module based on the PoE Hinton (2002), used to integrate the latent distributions generated by the dual-path encoders.

- A multi-task decoder that is simultaneously responsible for image reconstruction and the generation of gene expression profiles based on the Negative Binomial Distribution Lopez et al. (2018).

## C.2   Detailed Network Structure

### C.2.1   Image histology Encoder

This encoder is responsible for extracting rich, hierarchical visual features from the input histological image patches.

- **Input**: An image patch $I_i \in \mathbb{R}^{N \times H \times W \times C}$, where $H$ and $W$ are the height and width of a single spot's image patch, $C$ is the number of channels (for RGB images, $C = 3$), and $N$ is the total number of neighborhood patches.

- **Backbone Network**: We use a Swin Transformer Liu et al. (2021) as the feature extraction backbone. This network, through its hierarchical structure and shifted window self-attention mechanism Liu et al. (2021), can effectively capture long-range dependencies within the image, which is crucial for understanding complex tissue structures Dosovitskiy et al. (2020).

- **Feature Extraction**: The Swin Transformer maps the input image $I_i$ to a fixed-dimensional feature vector $f_{\text{img}} \in \mathbb{R}^{D_{\text{feat}}}$.

$$f_{\text{img}} = \text{SwinTransformer}(I_i) \tag{21}$$

- **Latent Space Mapping**: This feature vector $f_{\text{img}}$ is then passed through two independent fully connected (FC) layers Lopez et al. (2018) to generate the mean vector $\mu_{\text{img}} \in \mathbb{R}^{D_{\text{latent}}}$ and the log-variance vector $\log \sigma_{\text{img}}^2 \in \mathbb{R}^{D_{\text{latent}}}$ of the image modality latent space, respectively.

$$\mu_{\text{img}} = \text{FC}_{\mu,\text{img}}(f_{\text{img}}) \tag{22}$$

$$\log \sigma_{\text{img}}^2 = \text{FC}_{\sigma,\text{img}}(f_{\text{img}}) \tag{23}$$

To leverage the prior knowledge from large-scale natural image datasets, our Swin Transformer backbone loads weights pre-trained on the ImageNet dataset Deng et al. (2009).

### C.2.2   Spatial Context Encoder

This encoder is used to capture the global positional information of each image patch within the tissue slice.

- **Input**: A two-dimensional spatial coordinate vector $s_i = (x_i, y_i) \in \mathbb{R}^2$, representing the center coordinates of the image patch $I_i$.

- **Network Structure**: We use a Multilayer Perceptron [3] with two hidden layers to perform a non-linear transformation on the coordinate information. The activation function in the network is the Rectified Linear Unit (ReLU).

- **Feature Extraction**: The MLP maps the input coordinates $s_i$ to a high-dimensional feature vector $f_{\text{spatial}} \in \mathbb{R}^{D_{\text{hidden}}}$.

$$f_{\text{spatial}} = \text{MLP}_{\text{spatial}}(s_i) \tag{24}$$

- **Latent Space Mapping**: Similar to the image encoder, $f_{\text{spatial}}$ is passed through two independent fully connected layers to generate the mean vector $\mu_{\text{spatial}} \in \mathbb{R}^{D_{\text{latent}}}$ and the log-variance vector $\log \sigma_{\text{spatial}}^2 \in \mathbb{R}^{D_{\text{latent}}}$ of the spatial modality latent space.

$$\mu_{\text{spatial}} = \text{FC}_{\mu,\text{sp}}(f_{\text{spatial}}) \tag{25}$$

$$\log \sigma_{\text{spatial}}^2 = \text{FC}_{\sigma,\text{sp}}(f_{\text{spatial}}) \tag{26}$$

### C.2.3   Multimodal Fusion and Latent Space Sampling

- **PoE Fusion** To integrate information from the two modalities and their respective uncertainties, we adopt the PoE framework Hinton (2002). We sum the precisions (the inverse of the variance) of the latent distributions output by the two encoders (both are Gaussian distributions) to calculate the precision of the fused distribution, and then derive the fused mean and variance.

– **Precision Calculation**:

$$T_{\text{img}} = (\sigma_{\text{img}}^2)^{-1} = \exp(-\log \sigma_{\text{img}}^2) \tag{27}$$

$$T_{\text{spatial}} = (\sigma_{\text{spatial}}^2)^{-1} = \exp(-\log \sigma_{\text{spatial}}^2) \tag{28}$$

– **Fused Distribution Parameters**:

$$\sigma_{\text{fused}}^2 = (T_{\text{img}} + T_{\text{spatial}})^{-1} \tag{29}$$

$$\mu_{\text{fused}} = (\mu_{\text{img}} T_{\text{img}} + \mu_{\text{spatial}} T_{\text{spatial}}) \sigma_{\text{fused}}^2 \tag{30}$$

- **Reparameterization Sampling** To enable backpropagation of gradients through the sampling process, we use the reparameterization trick Kingma & Welling (2013). We sample a random noise vector $\epsilon \sim \mathcal{N}(0, I)$ from a standard normal distribution and then generate the final latent vector $z \in \mathbb{R}^{D_{\text{latent}}}$.

$$z = \mu_{\text{fused}} + \sigma_{\text{fused}} \odot \epsilon \tag{31}$$

where $\odot$ denotes element-wise multiplication.

## C.3 MULTI-TASK DECODER

### C.3.1 IMAGE RECONSTRUCTION DECODER

This decoder reconstructs the original histological image from the latent vector $z$.

- **Initial Transformation**: First, a fully connected layer projects $z$ into a high-dimensional space and reshapes it into a small 3D feature map $h_{\text{img}} \in \mathbb{R}^{C' \times H' \times W'}$ to serve as the starting point for subsequent convolutional operations.

- **Upsampling**: Next, a series of transposed convolution modules (including ConvTranspose2d Radford et al. (2015), BatchNorm2d Santurkar et al. (2018), and LeakyReLUMaas et al. (2013)) are used to progressively increase the spatial dimensions of the feature map while reducing its number of channels.

- **Final Output**: The final layer is a $3 \times 3$ convolutional layer that maps the feature map's channel count back to the number of channels of the input image, $C$. A Tanh activation function Chen et al. (2020) is then used to normalize the output pixel values to the range $[-1, 1]$, yielding the reconstructed image $I_{\text{recon}}$.

### C.3.2 GENE EXPRESSION DECODER

This decoder generates the distribution parameters for the gene expression profile from the latent vector $z$.

- **Feature Transformation**: The latent vector $z$ is first passed through an MLP network, which includes Batch Normalization (BatchNorm1d) Ioffe & Szegedy (2015), ReLU Agarap (2018), and Dropout Srivastava et al. (2014), to extract high-level features $h_{\text{rna}}$ for gene expression prediction.

- **Parameter Prediction**: The feature vector $h_{\text{rna}}$ is then fed into two independent linear output layers, which are used to predict the mean parameter $\mu_{\text{rna}} \in \mathbb{R}^M$ and the dispersion parameter $\theta_{\text{rna}} \in \mathbb{R}^M$ of the NB distribution, where $M$ is the number of target genes.

$$\mu_{\text{rna}} = \text{Softplus}(\text{Linear}_{\mu,\text{rna}}(h_{\text{rna}})) \tag{32}$$

$$\theta_{\text{rna}} = \text{Softplus}(\text{Linear}_{\theta,\text{rna}}(h_{\text{rna}})) \tag{33}$$

We use the Softplus activation function Baltrušaitis et al. (2018) to ensure that the values of $\mu_{\text{rna}}$ and $\theta_{\text{rna}}$ are positive, which is consistent with the parameter definition of the Negative Binomial distribution.

### C.4 LOSS FUNCTION AND OPTIMIZATION

#### C.4.1 COMPOSITE LOSS FUNCTION

The training objective of STevs is optimized through a carefully designed composite loss function $L_{\text{total}}$, which consists of four components:

$$L_{\text{total}} = \lambda_{\text{img}} L_{\text{img}} + \lambda_{\text{rna}} L_{\text{rna}} + \beta L_{\text{KLD}} + \gamma L_{\text{align}} \tag{34}$$

- **Image Reconstruction Loss ($L_{\text{img}}$)** We use the MSE to measure the difference between the reconstructed image $I_{\text{recon}}$ and the original image $I_{\text{true}}$:

$$L_{\text{img}} = \frac{1}{N} \sum_{i=1}^{N} \|I_{\text{recon}}(i) - I_{\text{true}}(i)\|_2^2 \tag{35}$$

- **Gene Expression Reconstruction Loss ($L_{\text{rna}}$)** We use the Negative Log-Likelihood (NLL) Lopez et al. (2018)of the Negative Binomial distribution. The probability mass function (PMF) of the Negative Binomial distribution is defined as:

$$P(Y = k|\mu, \theta) = \frac{\Gamma(k + \theta)}{\Gamma(k + 1)\Gamma(\theta)} \left(\frac{\theta}{\theta + \mu}\right)^{\theta} \left(\frac{\mu}{\theta + \mu}\right)^{k} \tag{36}$$

where $\Gamma(\cdot)$ is the Gamma function.

- **KL Divergence Loss ($L_{\text{KLD}}$)** As a standard component of the VAE framework Kingma & Welling (2013), we use the KL divergence to regularize the fused latent space. For a single sample, its analytic form is Cover (1999):

$$L_{\text{KLD}} = \frac{1}{2} \sum_{j=1}^{D_{\text{latent}}} (\sigma_{\text{fused},j}^2 + \mu_{\text{fused},j}^2 - 1 - \log \sigma_{\text{fused},j}^2) \tag{37}$$

- **Latent Space Alignment Loss ($L_{\text{align}}$)** We introduce an additional MSE loss to encourage different modality encoders to learn semantically aligned representations Radford et al. (2015); Wu & Goodman (2018). This loss directly pulls the mean vectors of the two modalities closer in the latent space.

#### C.4.2 OPTIMIZATION STRATEGY

- **KL Annealing** To stabilize the training of the VAE, we adopt a KL annealing strategy Bowman et al. (2016). The weight $\beta$ in the loss function is dynamically adjusted during training, its value increasing linearly from 0 to a preset maximum value $\beta_{\max}$ with training epoch $e$, and then remaining constant.

$$\beta_e = \beta_{\max} \cdot \min\left(1.0, \frac{e}{E_{\text{anneal}}}\right) \tag{38}$$

where $E_{\text{anneal}}$ is the total number of annealing epochs.

- **Optimizer** We use the AdamW optimizer Loshchilov & Hutter (2017) to update all learnable parameters of the model. Compared to the traditional Adam, AdamW typically provides better generalization performance by decoupling weight decay from the gradient update.

## D DATA PROCESSING AND EXPERIMENTAL DESIGN

### D.1 DATASETS AND PREPROCESSING

#### D.1.1 DATA SOURCES

Our study utilized a total of five dataset groups from two sources. The first three groups are from the public human dorsolateral prefrontal cortex (DLPFC) dataset Maynard et al. (2021), comprising 12 tissue sections from 3 different individuals. The latter two groups are public Visium mouse

brain datasets from the 10x Genomics platform Ståhl et al. (2016), containing 4 tissue sections from the same rmice but different egion. An overview of these datasets is provided in Figure 6 and Table 7. Additionally, to further validate the model's generalization ability, we conducted extended experiments on the Human Breast Cancer (HBC) Wu et al. (2021) and HSC Ji et al. (2020) datasets. These datasets is provided in Figure 22.

### D.1.2 IMAGE PREPROCESSING

For each spot, we extracted image patches from the corresponding high-resolution H&E stained whole-slide image (Figure 7). Specifically, we defined a perceptual field of a $5 \times 5$ grid of base patches centered on each spot's coordinates. With each base patch having a resolution of $32 \times 32$ pixels, this resulted in a final input image tensor of $160 \times 160 \times 3$ for the model, capturing both the fine-grained histology of the target spot and its adjacent microenvironment. Prior to being fed into the model, all image patches were normalized to the range $[-1, 1]$ using min-max normalization to stabilize training:

$$I_{\text{norm}} = \frac{2(I - I_{\min})}{I_{\max} - I_{\min}} - 1 \tag{39}$$

where $I$ is the original pixel value, and $I_{\min}$ and $I_{\max}$ are the minimum and maximum pixel values within the patch, respectively. The normalized patches also serve as the ground truth target for the image reconstruction task.

### D.1.3 SPATIAL COORDINATE PREPROCESSING

For each image patch, we extract the corresponding 2D coordinate vector $(x_i, y_i)$ from the tissue position file. These coordinates, representing the relative center position of the spot within the whole-slide image, are normalized and directly fed into the spatial encoder to preserve the global positional context of each patch. Generally, for relatively well-aligned slices, no operation on the spatial coordinates is necessary. However, for slices with discrepancies such as rotation or displacement, we recommend flattening the image patch corresponding to the coordinates to serve as features, and then using `Spateo` for coordinate alignment Qiu et al. (2024).

### D.1.4 GENE EXPRESSION PREPROCESSING

Given the high dimensionality and sparsity of SRT data, we performed a gene filtering step to identify SVGs. Using `scanpy` Wolf et al. (2018) and `squidpy` Palla et al. (2022), we calculated spatial autocorrelation (Moran's I) for each gene and retained those with a p-value less than 0.05. To ensure robustness, a gene was only included in the final set if it was identified as an SVG in at least two samples within the same dataset group. After filtering, each group contained over 2,000 SVGs. These filtered gene expression profiles serve as the ground truth target for the gene expression decoder, which models them using a Negative Binomial distribution to account for their count-based and over-dispersed nature.

## D.2 EXPERIMENTAL DESIGN AND EVALUATION

### D.2.1 EXPERIMENTAL SETTINGS

We evaluated model performance under two distinct settings:

- **Intra-slice Prediction:** For each slice, we randomly and independently split the spots into training, validation, and test sets with a 7:1:2 ratio to perform standard cross-validation within a single tissue slice.

- **Cross-slice Prediction:** To assess generalization, we employed a more challenging leave-one-out approach within each dataset group. One slice was used for training and all other slices in the group were used for testing (as illustrated in Fig. 2c). This setup mimics the real-world scenario of applying a pre-trained model to new, unseen patient samples.

### D.2.2 EVALUATION METRICS

To quantitatively evaluate the model's prediction accuracy for gene expression, we employed three standard statistical metrics Andersson et al. (2021); Asp et al. (2019):

- **Mean Squared Error (MSE)** Kingma & Welling (2013): To measure the average squared difference between predicted and actual gene expression values.
- **Pearson Correlation Coefficient (PCC)** Pearson (1896): To assess the linear relationship between predicted and ground-truth expression profiles.
- **Spearman's Rank Correlation Coefficient (SCC)** Spearman (1987): To evaluate the monotonic relationship, which is robust to outliers.

### D.2.3 EXPERIMENTAL SETUP

**Hardware Configuration** Our experiments were conducted on a high-performance server equipped with four NVIDIA A100 GPUs (80GB of VRAM each), dual Intel(R) Xeon(R) Gold 6267C CPUs, and 1.5TB of system memory. The runtimes reported in the main paper are for model inference on a single GPU.

**Training Parameters** We trained all models for 100 epochs using a learning rate of $1 \times 10^{-4}$ on four NVIDIA A100 (80GB) GPUs. A comprehensive list of all training hyperparameters is provided in Appendix J.

**Loss Function Weights** In our experiments, the default weights for the composite loss function were set to $\lambda_{\mathrm{img}} = 1.0$, $\lambda_{\mathrm{rna}} = 10.0$, $\beta = 0.5$, and $\gamma = 0.5$. A detailed sensitivity analysis of these weights on model performance is discussed in Appendix I.

### D.2.4 BASELINE SET

For the baseline models, we strictly adhered to their officially provided pipelines for training and evaluation, making minor adaptive modifications to some for compatibility. Notably, since iStar's methodology involves predicting all spots at once, we employed a masking strategy during the intra-slice training phase to conceal the test set samples and prevent data leakage. For the STAGE model, we deviated from its provided pipeline, which calculates metrics on the combined training and test sets. To maintain consistency and ensure a fair comparison with all other methods, we adopted a stricter approach, evaluating its performance exclusively on the test set.

## E   MAIN PERFORMANCE DETAILS

### E.1   DLPFC ON INTRA-SLICE EXPERIMENTS

As shown in Table 9, in the intra-slice experiments conducted on the DLPFC datasets, we systematically evaluated the performance of our STevs model against various existing mainstream methods. This evaluation was performed on three public datasets: Human 1, Human 2, and Human 3. The evaluation metrics include MSE, PCC, and SCC. The experimental results clearly indicate that our STevs model achieves optimal performance across all three datasets and on all three evaluation metrics. This data provides strong evidence for the superiority and robustness of the STevs model in the task of intra-slice spatial gene expression prediction.

### E.2   DLPFC ON CROSS-SLICE EXPERIMENTS

To further evaluate the model's generalization ability, this subsection presents the results from the more challenging cross-slice experiments. In this setting, the model must utilize information from the training slices to predict the gene expression profile of a completely unseen slice from the same tissue, posing a stringent test of its knowledge transfer capabilities. As detailed in Table 9, the performance comparison on the DLPFC datasets (Human 1, 2, and 3) shows that most baseline models suffered a significant performance drop due to their inability to generalize effectively. The predictions of some models were even indistinguishable from random noise (with PCC/SCC values close to

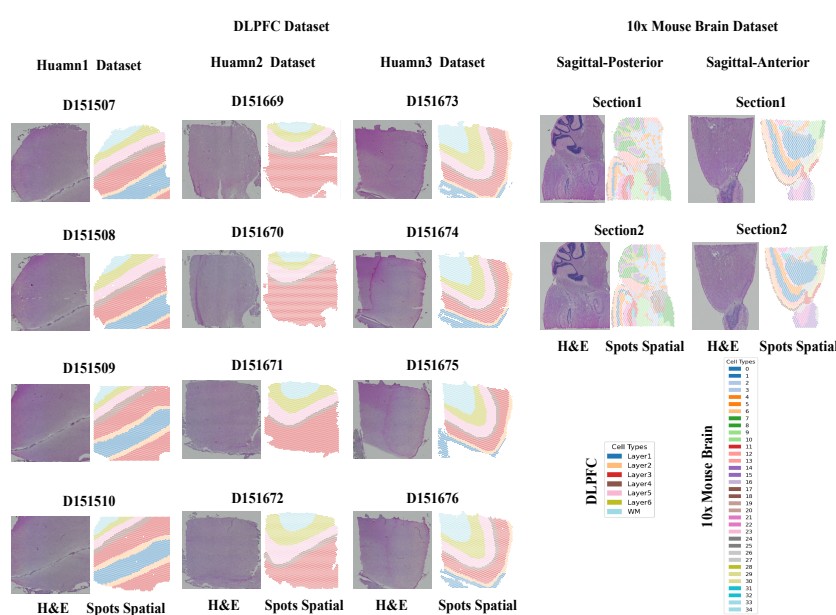

Figure 6: Overview of the primary spatial transcriptomics datasets used in this study. The figure shows the H&E images and their corresponding annotated cell type/tissue layer maps for a total of 16 datasets from five groups.

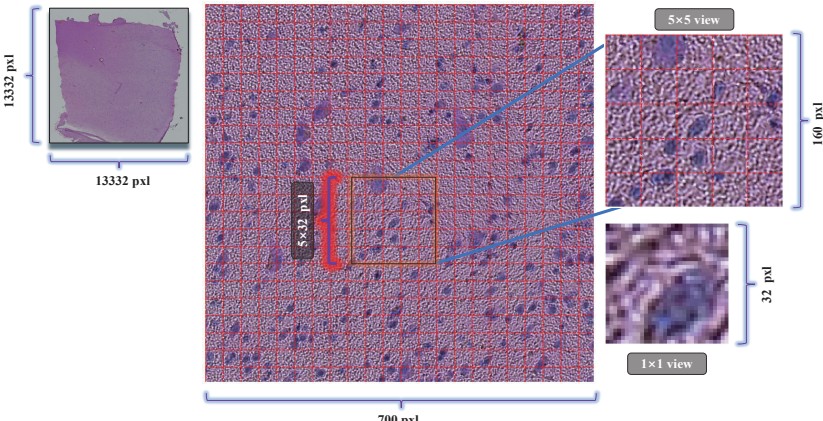

Figure 7: Schematic diagram of the image patch extraction process from a whole-slide H&E image

zero or negative). However, the STevs model still performed exceptionally well in this rigorous test, with its performance significantly surpassing all competing methods across all metrics. Specifically, compared to the best-performing baseline model, iStar, STevs demonstrated a substantial advantage in correlation metrics (PCC and SCC). For example, on the Human 3 dataset, the PCC of STevs reached 0.256, far exceeding iStar's 0.122. This result provides strong evidence for the powerful cross-slice generalization ability of the STevs model, showcasing its capacity to effectively transfer spatial pattern knowledge learned from training slices to new, unseen target slices.

### E.3 10X MOUSE BRAIN ON INTRA-SLICE EXPERIMENTS

To further validate our model's broad applicability and cross-species generalization ability, we also conducted a series of rigorous intra-slice performance evaluations on the 10x Mouse Brain dataset. As shown in Table 10, we performed a comprehensive performance comparison between STevs and various mainstream baseline models on two different brain region slices: Sagittal-Anterior and

Table 7: Spots and Gene Counts per Sample Group

| Group | Sample | Spots Number | Gene Number |
|---|---|---|---|
| Human 1 | DLPFC 151507 | 4221 | 2239 |
| | DLPFC 151508 | 4381 | 2239 |
| | DLPFC 151509 | 4788 | 2239 |
| | DLPFC 151510 | 4595 | 2239 |
| Human 2 | DLPFC 151669 | 3636 | 2253 |
| | DLPFC 151670 | 3484 | 2253 |
| | DLPFC 151671 | 4093 | 2253 |
| | DLPFC 151672 | 3888 | 2253 |
| Human 3 | DLPFC 151673 | 3611 | 3271 |
| | DLPFC 151674 | 3635 | 3271 |
| | DLPFC 151675 | 3566 | 3271 |
| | DLPFC 151676 | 3431 | 3271 |
| Sagittal-Anterior | Sagittal-Anterior section1 (SA-1) | 2695 | 3310 |
| | Sagittal-Anterior section2 (SA-2) | 2825 | 3310 |
| Sagittal-Posterior | Sagittal-Posterior section1 (SP-1) | 3355 | 3310 |
| | Sagittal-Posterior section2 (SP-2) | 3289 | 3310 |

Table 8: Model Performance on Human 1, Human 2, and Human 3 Datasets

| Model Category | Human 1 | | | Human 2 | | | Human 3 | | |
|---|---|---|---|---|---|---|---|---|---|
| | MSE $\downarrow$ | PCC $\uparrow$ | SCC $\uparrow$ | MSE $\downarrow$ | PCC $\uparrow$ | SCC $\uparrow$ | MSE $\downarrow$ | PCC $\uparrow$ | SCC $\uparrow$ |
| **Local Image-based** | | | | | | | | | |
| ST-Net (Nat. B.E. He et al. (2020)) | 1.494 ± 0.048 | 0.033 ± 0.022 | 0.070 ± 0.011 | 1.348 ± 0.028 | 0.053 ± 0.023 | 0.086 ± 0.020 | 0.365 ± 0.068 | 0.123 ± 0.019 | 0.134 ± 0.045 |
| BLEEP (NeurIPS Xie et al. (2023)) | 1.551 ± 0.058 | 0.036 ± 0.003 | 0.037 ± 0.005 | 1.365 ± 0.067 | 0.058 ± 0.011 | 0.057 ± 0.013 | 1.067 ± 0.085 | 0.086 ± 0.007 | 0.077 ± 0.008 |
| **Graph-based Context** | | | | | | | | | |
| EGN (PR Yang et al. (2024)) | 0.995 ± 0.053 | 0.051 ± 0.006 | 0.054 ± 0.005 | 1.008 ± 0.035 | 0.053 ± 0.006 | 0.066 ± 0.011 | 0.997 ± 0.074 | 0.103 ± 0.016 | 0.109 ± 0.014 |
| IGI-DL (Cell R.M. Gao et al. (2024)) | 0.205 ± 0.009 | 0.115 ± 0.008 | 0.117 ± 0.008 | 0.297 ± 0.023 | 0.155 ± 0.048 | 0.152 ± 0.048 | 0.284 ± 0.048 | 0.138 ± 0.036 | 0.124 ± 0.026 |
| **Transformer-based Context** | | | | | | | | | |
| iStar (Nat. Biot. Zhang et al. (2024)) | 0.149 ± 0.050 | 0.191 ± 0.029 | 0.189 ± 0.016 | 0.194 ± 0.058 | 0.204 ± 0.016 | 0.229 ± 0.008 | 0.171 ± 0.036 | 0.236 ± 0.020 | 0.230 ± 0.018 |
| TRIPLEX (CVPR Chung et al. (2024)) | 0.181 ± 0.009 | 0.131 ± 0.007 | 0.125 ± 0.007 | 0.211 ± 0.006 | 0.194 ± 0.012 | 0.186 ± 0.013 | 0.179 ± 0.014 | 0.211 ± 0.020 | 0.199 ± 0.020 |
| M2ORT (AAAI Wang et al. (2025)) | 1.000 ± 0.003 | -0.001 ± 0.001 | 0.000 ± 0.002 | 1.006 ± 0.006 | -0.001 ± 0.001 | -0.000 ± 0.001 | 1.019 ± 0.010 | -0.000 ± 0.001 | -0.000 ± 0.002 |
| **Coordinate-based Generative** | | | | | | | | | |
| STAGE (NAR Li et al. (2024)) | 0.259 ± 0.007 | 0.108 ± 0.013 | 0.105 ± 0.017 | 0.307 ± 0.018 | 0.139 ± 0.034 | 0.130 ± 0.029 | 0.339 ± 0.046 | 0.150 ± 0.016 | 0.149 ± 0.013 |
| STevs (Ours) | **0.142 ± 0.008** | **0.215 ± 0.011** | **0.202 ± 0.008** | **0.188 ± 0.005** | **0.281 ± 0.005** | **0.271 ± 0.004** | **0.166 ± 0.014** | **0.296 ± 0.019** | **0.263 ± 0.020** |

Sagittal-Posterior. The experimental results clearly show that the STevs model consistently outperformed all competing methods on both mouse brain datasets. This successful validation on datasets from a different species provides strong evidence for the STevs model's powerful generalization ability and its great potential for application as a general-purpose framework in diverse biological scenarios.

### E.4 10X MOUSE BRAIN ON CROSS-SLICE EXPERIMENTS

In this section, we subject our model to the most rigorous test: a cross-slice generalization performance evaluation on the 10x Mouse Brain dataset. This task requires the model to transfer and apply knowledge learned from one brain region slice to another, completely unseen one, posing the ultimate challenge to its generalization and robustness. The experimental results in Table 11 once again unequivocally demonstrate the superior performance of STevs. In this highly challenging scenario, STevs not only far surpassed most baseline models, whose predictions were close to random, but also achieved a comprehensive and significant victory over the strongest competitor, iStar. This is especially evident in the prediction for the posterior (Sagittal-Posterior) slice, where STevs's PCC reached as high as 0.442, a substantial lead compared to iStar's 0.363, fully reflecting its powerful predictive capability and generalization stability. Synthesizing the performance on both human and mouse datasets, these cross-slice experimental results ultimately establish the status of STevs as a high-performance, cross-species applicable, and general-purpose framework for spatial gene expression prediction.

Table 9: Model Performance on Human 1, Human 2, and Human 3 Datasets

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

### F.1    CONTRIBUTION OF CORE COMPONENTS

The experimental results clearly reveal the contribution of each core component. Removing the **Spatial Encoder** (`STevs w/o Spatial Encoder`) leads to a significant performance drop in the cross-slice task, which demonstrates that relying solely on a powerful visual feature extractor is insufficient; spatial coordinate information is crucial for capturing the macroscopic patterns of gene expression. Removing the **Latent Space Alignment Loss** (`STevs w/o Alignment Loss`) has a minor impact on the simpler intra-slice task but leads to performance degradation in the cross-slice setting. This indicates that the alignment loss is effective for learning a slice-invariant spatial representation. Finally, removing the **Image Decoder** (`STevs w/o Image Decoder`) causes a slight but consistent decrease in performance, proving that image reconstruction serves as an important auxiliary task and regularizer, compelling the encoder to learn more fine-grained visual representations.

### F.2    EFFECTIVENESS OF THE MULTIMODAL FUSION MECHANISM

Our probabilistic PoE fusion mechanism is significantly superior to other common fusion methods. Simple **Feature Concatenation** (`STevs (Concat)`) and **Deterministic Mean Fusion** (`STevs (Deterministic)`) perform far below our model, especially in the cross-slice task, as they cannot effectively handle the uncertainty and relative importance of different modalities. Interestingly, the more advanced **Cross-Attention** mechanism (`STevs (Cross-Attention)`), while showing competitive performance on some intra-slice tasks, exhibits insufficient generalization ability with a noticeable performance drop in the cross-slice setting. This, in turn, highlights the superiority of our PoE-based probabilistic fusion method in modeling uncertainty and enhancing generalization.

Table 11: Cross-slice Performance Comparison on 10x Mouse Brain Datasets

| Model Category | Sagittal-Anterior | | | Sagittal-Posterior | | |
|---|---|---|---|---|---|---|
| | MSE ↓ | PCC ↑ | SCC ↑ | MSE ↓ | PCC ↑ | SCC ↑ |
| **Local Image-based** | | | | | | |
| ST-Net (Nat. B.E. He et al. (2020)) | $1.861 \pm 0.084$ | $0.010 \pm 0.008$ | $0.052 \pm 0.010$ | $1.502 \pm 0.020$ | $0.071 \pm 0.022$ | $0.131 \pm 0.002$ |
| BLEEP (NeurIPS Xie et al. (2023)) | $1.436 \pm 0.027$ | $0.069 \pm 0.003$ | $0.067 \pm 0.002$ | $1.229 \pm 0.009$ | $0.118 \pm 0.024$ | $0.111 \pm 0.024$ |
| **Graph-based Context** | | | | | | |
| EGN (PR Yang et al. (2024)) | $1.159 \pm 0.055$ | $0.084 \pm 0.006$ | $0.075 \pm 0.004$ | $0.825 \pm 0.014$ | $0.117 \pm 0.000$ | $0.130 \pm 0.002$ |
| IGI-DL (Cell R.M. Gao et al. (2024)) | $0.918 \pm 0.071$ | $0.089 \pm 0.005$ | $0.087 \pm 0.007$ | $0.924 \pm 0.024$ | $0.118 \pm 0.037$ | $0.126 \pm 0.030$ |
| **Transformer-based Context** | | | | | | |
| iStar (Nat. Biot. Zhang et al. (2024)) | $\underline{0.273 \pm 0.070}$ | $\underline{0.301 \pm 0.014}$ | $\underline{0.300 \pm 0.017}$ | $\underline{0.269 \pm 0.105}$ | $\underline{0.363 \pm 0.038}$ | $\underline{0.325 \pm 0.030}$ |
| TRIPLEX (CVPR Chung et al. (2024)) | $0.438 \pm 0.022$ | $0.197 \pm 0.010$ | $0.180 \pm 0.013$ | $0.450 \pm 0.008$ | $0.256 \pm 0.006$ | $0.247 \pm 0.006$ |
| M2ORT (AAAI Wang et al. (2025)) | $1.133 \pm 0.000$ | $0.006 \pm 0.002$ | $0.007 \pm 0.003$ | $1.253 \pm 0.145$ | $0.001 \pm 0.006$ | $0.001 \pm 0.005$ |
| **Coordinate-based Generative** | | | | | | |
| STAGE (NAR Li et al. (2024)) | $0.624 \pm 0.028$ | $0.125 \pm 0.021$ | $0.118 \pm 0.019$ | $0.631 \pm 0.021$ | $0.156 \pm 0.028$ | $0.158 \pm 0.026$ |
| STevs (Ours) | $\mathbf{0.261 \pm 0.011}$ | $\mathbf{0.362 \pm 0.014}$ | $\mathbf{0.351 \pm 0.018}$ | $\mathbf{0.223 \pm 0.003}$ | $\mathbf{0.442 \pm 0.000}$ | $\mathbf{0.392 \pm 0.006}$ |

## F.3 CHOICE OF ENCODER ARCHITECTURES

Architectural comparisons validate the rationale of our choices. For the image encoder, the **Swin Transformer**, with its hierarchical structure, outperforms both a standard `ViT` and a traditional `CNN`. Furthermore, removing the **ImageNet pre-trained weights** (`STevs w/o Pretrained`) leads to a substantial drop in performance, demonstrating the effectiveness of transfer learning. For the spatial encoder, we also explored more complex alternatives, including a **Gaussian Process** (`STevs (Gaussian Process)`) and an **MLP with Fourier features** (`STevs (MLP w/ Fourier)`). Although these variants perform adequately on the intra-slice task, their performance degrades severely in the cross-slice setting. This indicates that our simple MLP, when combined with our fusion and alignment strategy, provides a more robust and generalizable foundation.

In summary, this series of exhaustive ablation studies, spanning different species and task difficulties, systematically validates the necessity and advanced nature of each design element in the STevs model, collectively forming the solid foundation for its accurate and robust predictions in diverse biological scenarios. Additionally, the decoder could be modified into a Transformer decoder to consider gene co-expression and potentially improve prediction performance Cui et al. (2024); however, this is beyond the scope of this paper's focus on representation fusion.

Table 12: Intra-slice cross-validation Performance Comparison of STevs Variants on DLPFC Datasets

| Model Variant | Human 1 | | | Human 2 | | | Human 3 | | |
|---|---|---|---|---|---|---|---|---|---|
| | MSE ↓ | PCC ↑ | SCC ↑ | MSE ↓ | PCC ↑ | SCC ↑ | MSE ↓ | PCC ↑ | SCC ↑ |
| *Component Ablation* | | | | | | | | | |
| STevs w/o Image Decoder | $0.151 \pm 0.010$ | $0.203 \pm 0.015$ | $0.195 \pm 0.016$ | $0.192 \pm 0.012$ | $0.274 \pm 0.018$ | $0.260 \pm 0.019$ | $0.176 \pm 0.018$ | $0.283 \pm 0.021$ | $0.253 \pm 0.023$ |
| STevs w/o Spatial Encoder | $0.171 \pm 0.025$ | $0.172 \pm 0.021$ | $0.162 \pm 0.022$ | $0.226 \pm 0.031$ | $0.225 \pm 0.025$ | $0.217 \pm 0.026$ | $0.199 \pm 0.032$ | $0.237 \pm 0.028$ | $0.210 \pm 0.030$ |
| STevs w/o Alignment Loss | $0.147 \pm 0.009$ | $0.209 \pm 0.012$ | $0.200 \pm 0.013$ | $0.188 \pm 0.010$ | $0.280 \pm 0.015$ | $0.266 \pm 0.016$ | $0.172 \pm 0.015$ | $0.289 \pm 0.019$ | $0.261 \pm 0.020$ |
| *Fusion Mechanism Ablation* | | | | | | | | | |
| STevs (Concat) | $0.171 \pm 0.018$ | $0.191 \pm 0.017$ | $0.179 \pm 0.019$ | $0.226 \pm 0.021$ | $0.243 \pm 0.024$ | $0.239 \pm 0.025$ | $0.201 \pm 0.022$ | $0.260 \pm 0.027$ | $0.222 \pm 0.028$ |
| STevs (Deterministic) | $0.184 \pm 0.015$ | $0.179 \pm 0.014$ | $0.177 \pm 0.015$ | $0.241 \pm 0.018$ | $0.234 \pm 0.020$ | $0.231 \pm 0.021$ | $0.213 \pm 0.019$ | $0.251 \pm 0.023$ | $0.215 \pm 0.024$ |
| STevs (Cross-Attention) | $0.143 \pm 0.014$ | $0.213 \pm 0.019$ | $0.201 \pm 0.018$ | $0.187 \pm 0.015$ | $0.283 \pm 0.021$ | $0.273 \pm 0.020$ | $0.167 \pm 0.021$ | $0.294 \pm 0.026$ | $0.261 \pm 0.027$ |
| *Spatial Encoder Variants* | | | | | | | | | |
| STevs (Gaussian Process) | $0.144 \pm 0.010$ | $0.212 \pm 0.013$ | $0.200 \pm 0.014$ | $0.189 \pm 0.011$ | $0.279 \pm 0.016$ | $0.269 \pm 0.017$ | $0.168 \pm 0.016$ | $0.293 \pm 0.020$ | $0.260 \pm 0.021$ |
| STevs (MLP w/ Fourier) | $0.145 \pm 0.011$ | $0.210 \pm 0.014$ | $0.198 \pm 0.015$ | $0.191 \pm 0.012$ | $0.276 \pm 0.017$ | $0.265 \pm 0.018$ | $0.170 \pm 0.017$ | $0.290 \pm 0.021$ | $0.258 \pm 0.022$ |
| *Architecture Variants* | | | | | | | | | |
| STevs (Convolutional) | $0.217 \pm 0.022$ | $0.163 \pm 0.018$ | $0.162 \pm 0.019$ | $0.259 \pm 0.026$ | $0.215 \pm 0.028$ | $0.203 \pm 0.029$ | $0.246 \pm 0.028$ | $0.224 \pm 0.031$ | $0.203 \pm 0.033$ |
| STevs (ViT) | $0.149 \pm 0.012$ | $0.210 \pm 0.015$ | $0.194 \pm 0.014$ | $0.199 \pm 0.015$ | $0.271 \pm 0.018$ | $0.266 \pm 0.017$ | $0.176 \pm 0.018$ | $0.290 \pm 0.023$ | $0.252 \pm 0.025$ |
| STevs w/o Pretrained | $0.191 \pm 0.026$ | $0.176 \pm 0.025$ | $0.173 \pm 0.024$ | $0.243 \pm 0.030$ | $0.231 \pm 0.033$ | $0.218 \pm 0.032$ | $0.223 \pm 0.033$ | $0.239 \pm 0.037$ | $0.213 \pm 0.036$ |
| **STevs (Ours)** | $0.142 \pm 0.013$ | $0.215 \pm 0.018$ | $0.202 \pm 0.017$ | $0.188 \pm 0.014$ | $0.281 \pm 0.020$ | $0.271 \pm 0.019$ | $0.166 \pm 0.020$ | $0.296 \pm 0.025$ | $0.263 \pm 0.026$ |

## G GENE EXPRESSION VISUALIZATION FOR EACH DATASET

This appendix section provides supplementary visualizations for the gene expression prediction performance of STevs and all baseline models, corresponding to the results discussed in the main manuscript. The following figures are organized by the two core validation strategies.

**Intra-Slice Validation** Figures 8-12 display the qualitative results for the intra-slice validation task. For these experiments, models were evaluated on a 20% held-out test set from within each individual slice. Visualizations are shown for representative spatially variable genes: OLFM1 Shen

Table 13: Cross-slice cross-validation Performance Comparison of STevs Variants on DLPFC Datasets

| Model Variant | Human 1 | | | Human 2 | | | Human 3 | | |
|---|---|---|---|---|---|---|---|---|---|
| | MSE ↓ | PCC ↑ | SCC ↑ | MSE ↓ | PCC ↑ | SCC ↑ | MSE ↓ | PCC ↑ | SCC ↑ |
| *Component Ablation* | | | | | | | | | |
| STevs w/o Image Decoder | 0.160 ± 0.018 | 0.143 ± 0.025 | 0.138 ± 0.024 | 0.227 ± 0.035 | 0.148 ± 0.041 | 0.150 ± 0.040 | 0.196 ± 0.033 | 0.233 ± 0.040 | 0.203 ± 0.038 |
| STevs w/o Spatial Encoder | 0.350 ± 0.031 | 0.103 ± 0.015 | 0.113 ± 0.016 | 0.324 ± 0.033 | 0.144 ± 0.020 | 0.137 ± 0.021 | 0.357 ± 0.038 | 0.156 ± 0.023 | 0.153 ± 0.024 |
| STevs w/o Alignment Loss | 0.158 ± 0.017 | 0.145 ± 0.024 | 0.142 ± 0.023 | 0.225 ± 0.036 | 0.150 ± 0.042 | 0.148 ± 0.041 | 0.195 ± 0.032 | 0.235 ± 0.039 | 0.205 ± 0.037 |
| *Fusion Mechanism Ablation* | | | | | | | | | |
| STevs (Concat) | 0.211 ± 0.022 | 0.117 ± 0.019 | 0.110 ± 0.018 | 0.290 ± 0.029 | 0.122 ± 0.025 | 0.122 ± 0.026 | 0.252 ± 0.031 | 0.190 ± 0.028 | 0.165 ± 0.027 |
| STevs (Deterministic) | 0.233 ± 0.019 | 0.107 ± 0.016 | 0.108 ± 0.017 | 0.303 ± 0.024 | 0.120 ± 0.022 | 0.111 ± 0.023 | 0.278 ± 0.026 | 0.173 ± 0.025 | 0.157 ± 0.024 |
| STevs (Cross-Attention) | 0.155 ± 0.020 | 0.148 ± 0.027 | 0.145 ± 0.026 | 0.242 ± 0.038 | 0.134 ± 0.044 | 0.133 ± 0.043 | 0.255 ± 0.041 | 0.128 ± 0.045 | 0.120 ± 0.046 |
| *Spatial Encoder Variants* | | | | | | | | | |
| STevs (Gaussian Process) | 0.345 ± 0.030 | 0.108 ± 0.016 | 0.117 ± 0.017 | 0.318 ± 0.032 | 0.149 ± 0.021 | 0.141 ± 0.022 | 0.352 ± 0.037 | 0.160 ± 0.024 | 0.156 ± 0.025 |
| STevs (MLP w/ Fourier) | 0.341 ± 0.029 | 0.112 ± 0.017 | 0.120 ± 0.018 | 0.315 ± 0.031 | 0.152 ± 0.022 | 0.145 ± 0.023 | 0.349 ± 0.036 | 0.163 ± 0.025 | 0.159 ± 0.026 |
| *Architecture Variants* | | | | | | | | | |
| STevs (Convolutional) | 0.298 ± 0.032 | 0.089 ± 0.014 | 0.090 ± 0.013 | 0.362 ± 0.038 | 0.104 ± 0.018 | 0.095 ± 0.019 | 0.343 ± 0.040 | 0.132 ± 0.021 | 0.124 ± 0.022 |
| STevs (ViT) | 0.166 ± 0.019 | 0.140 ± 0.026 | 0.134 ± 0.025 | 0.237 ± 0.037 | 0.145 ± 0.043 | 0.147 ± 0.042 | 0.201 ± 0.034 | 0.229 ± 0.041 | 0.200 ± 0.039 |
| STevs w/o Pretrained | 0.254 ± 0.036 | 0.099 ± 0.035 | 0.101 ± 0.034 | 0.319 ± 0.041 | 0.112 ± 0.048 | 0.104 ± 0.047 | 0.300 ± 0.046 | 0.159 ± 0.053 | 0.147 ± 0.052 |
| **STevs (Ours)** | **0.145 ± 0.021** | **0.153 ± 0.028** | **0.152 ± 0.027** | **0.202 ± 0.040** | **0.167 ± 0.045** | **0.166 ± 0.044** | **0.174 ± 0.038** | **0.256 ± 0.042** | **0.231 ± 0.040** |

Table 14: Intra-slice cross-validation Performance Comparison of STevs Variants on 10x Mouse Brain Datasets

| Model Variant | Sagittal-Anterior | | | Sagittal-Posterior | | |
|---|---|---|---|---|---|---|
| | MSE ↓ | PCC ↑ | SCC ↑ | MSE ↓ | PCC ↑ | SCC ↑ |
| *Component Ablation* | | | | | | |
| STevs w/o Image Decoder | 0.249 ± 0.018 | 0.403 ± 0.028 | 0.387 ± 0.030 | 0.215 ± 0.013 | 0.473 ± 0.016 | 0.415 ± 0.015 |
| STevs w/o Spatial Encoder | 0.330 ± 0.024 | 0.317 ± 0.021 | 0.287 ± 0.022 | 0.250 ± 0.020 | 0.389 ± 0.018 | 0.338 ± 0.019 |
| STevs w/o Alignment Loss | 0.241 ± 0.017 | 0.411 ± 0.027 | 0.390 ± 0.029 | 0.213 ± 0.012 | 0.479 ± 0.015 | 0.421 ± 0.014 |
| *Fusion Mechanism Ablation* | | | | | | |
| STevs (Concat) | 0.353 ± 0.025 | 0.300 ± 0.022 | 0.362 ± 0.026 | 0.262 ± 0.021 | 0.428 ± 0.023 | 0.373 ± 0.022 |
| STevs (Deterministic) | 0.353 ± 0.020 | 0.336 ± 0.019 | 0.310 ± 0.020 | 0.281 ± 0.017 | 0.411 ± 0.019 | 0.363 ± 0.018 |
| STevs (Cross-Attention) | 0.240 ± 0.022 | 0.411 ± 0.033 | 0.394 ± 0.034 | 0.206 ± 0.017 | 0.488 ± 0.021 | 0.425 ± 0.020 |
| *Spatial Encoder Variants* | | | | | | |
| STevs (Gaussian Process) | 0.240 ± 0.016 | 0.410 ± 0.026 | 0.393 ± 0.028 | 0.209 ± 0.011 | 0.483 ± 0.014 | 0.420 ± 0.013 |
| STevs (MLP w/ Fourier) | 0.238 ± 0.016 | 0.415 ± 0.027 | 0.399 ± 0.029 | 0.211 ± 0.012 | 0.480 ± 0.015 | 0.417 ± 0.014 |
| *Architecture Variants* | | | | | | |
| STevs (Convolutional) | 0.351 ± 0.028 | 0.307 ± 0.025 | 0.322 ± 0.027 | 0.332 ± 0.025 | 0.382 ± 0.026 | 0.313 ± 0.024 |
| STevs (ViT) | 0.251 ± 0.019 | 0.403 ± 0.029 | 0.391 ± 0.031 | 0.217 ± 0.014 | 0.476 ± 0.017 | 0.414 ± 0.016 |
| STevs w/o Pretrained | 0.329 ± 0.034 | 0.347 ± 0.036 | 0.318 ± 0.035 | 0.290 ± 0.031 | 0.403 ± 0.033 | 0.357 ± 0.032 |
| **STevs (Ours)** | **0.239 ± 0.021** | **0.413 ± 0.032** | **0.396 ± 0.033** | **0.208 ± 0.016** | **0.486 ± 0.020** | **0.423 ± 0.019** |

Table 15: Cross-slice cross-validation Performance Comparison of STevs Variants on 10x Mouse Brain Datasets

| Model Variant | Sagittal-Anterior | | | Sagittal-Posterior | | |
|---|---|---|---|---|---|---|
| | MSE ↓ | PCC ↑ | SCC ↑ | MSE ↓ | PCC ↑ | SCC ↑ |
| *Component Ablation* | | | | | | |
| STevs w/o Image Decoder | 0.291 ± 0.026 | 0.327 ± 0.029 | 0.309 ± 0.030 | 0.252 ± 0.020 | 0.403 ± 0.021 | 0.356 ± 0.022 |
| STevs w/o Spatial Encoder | 0.433 ± 0.035 | 0.212 ± 0.024 | 0.209 ± 0.025 | 0.351 ± 0.031 | 0.280 ± 0.026 | 0.263 ± 0.027 |
| STevs w/o Alignment Loss | 0.285 ± 0.027 | 0.331 ± 0.030 | 0.315 ± 0.031 | 0.249 ± 0.021 | 0.408 ± 0.022 | 0.360 ± 0.023 |
| *Fusion Mechanism Ablation* | | | | | | |
| STevs (Concat) | 0.372 ± 0.031 | 0.266 ± 0.028 | 0.261 ± 0.029 | 0.319 ± 0.026 | 0.356 ± 0.027 | 0.308 ± 0.028 |
| STevs (Deterministic) | 0.353 ± 0.027 | 0.251 ± 0.025 | 0.230 ± 0.026 | 0.347 ± 0.024 | 0.337 ± 0.025 | 0.400 ± 0.026 |
| STevs (Cross-Attention) | 0.313 ± 0.030 | 0.290 ± 0.028 | 0.280 ± 0.029 | 0.268 ± 0.024 | 0.354 ± 0.026 | 0.314 ± 0.027 |
| *Spatial Encoder Variants* | | | | | | |
| STevs (Gaussian Process) | 0.425 ± 0.034 | 0.218 ± 0.023 | 0.214 ± 0.024 | 0.345 ± 0.030 | 0.287 ± 0.025 | 0.270 ± 0.026 |
| STevs (MLP w/ Fourier) | 0.421 ± 0.033 | 0.223 ± 0.024 | 0.219 ± 0.025 | 0.340 ± 0.029 | 0.291 ± 0.026 | 0.275 ± 0.027 |
| *Architecture Variants* | | | | | | |
| STevs (Convolutional) | 0.450 ± 0.041 | 0.207 ± 0.031 | 0.196 ± 0.033 | 0.410 ± 0.038 | 0.279 ± 0.029 | 0.257 ± 0.030 |
| STevs (ViT) | 0.318 ± 0.029 | 0.301 ± 0.027 | 0.309 ± 0.028 | 0.261 ± 0.023 | 0.394 ± 0.020 | 0.356 ± 0.022 |
| STevs w/o Pretrained | 0.413 ± 0.052 | 0.237 ± 0.043 | 0.226 ± 0.044 | 0.370 ± 0.040 | 0.319 ± 0.036 | 0.286 ± 0.037 |
| **STevs (Ours)** | **0.261 ± 0.033** | **0.362 ± 0.035** | **0.350 ± 0.036** | **0.223 ± 0.028** | **0.442 ± 0.030** | **0.392 ± 0.029** |

et al. (2025) for the `human1`, `human2`, and `human3` datasets, and Dgkz Ishisaka & Hara (2014) for the `anterior` and `posterior` datasets.

**Cross-Slice Validation**  Figures 13 to 19 present the results for the more challenging cross-slice validation task. A model is trained on a single slice from a group (e.g., D151507) and evaluated on all other unseen slices from the same group. These figures visually demonstrate the robust generalization capability of our model in contrast to the baseline methods.

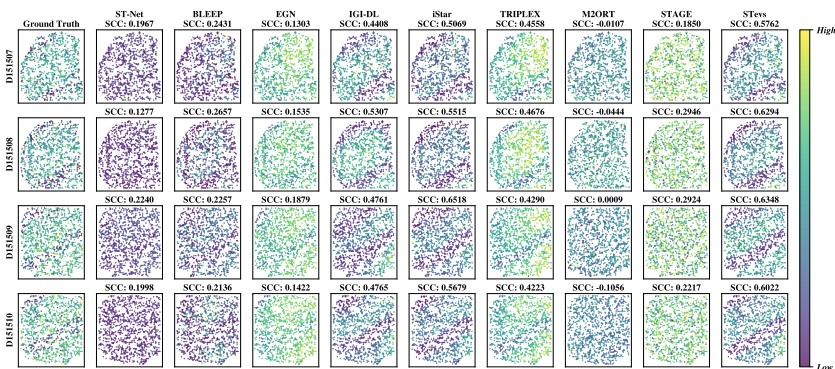

Figure 8: The results of intra-slice validation for the OLFM1 gene on 4 datasets of human1 (with a 20% test set)

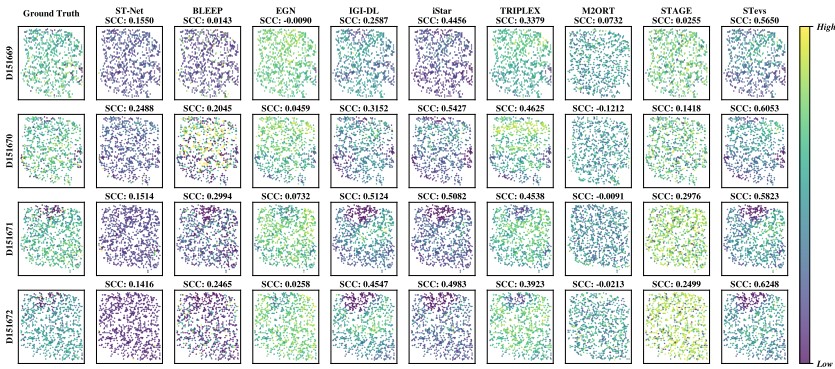

Figure 9: The results of intra-slice validation for the OLFM1 gene on 4 datasets of human2 (with a 20% test set)

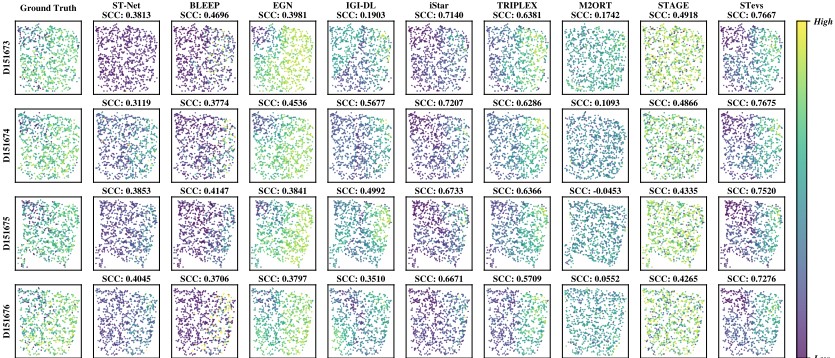

Figure 10: The results of intra-slice validation for the OLFM1 gene on 4 datasets of human3 (with a 20% test set)

## H  ACCURATE RECOVERY OF SPATIAL DOMAINS

To evaluate whether the gene expression profiles predicted by our model can accurately recover the spatial domains of the tissue, we performed clustering analysesMacQueen (1967) on the expression profiles generated by each method after cross-slice training, and calculated the ARI against the manually annotated ground-truth domains. As summarized in Table 16, STevs demonstrates superior performance across all five dataset groups. Notably, the ARI score from clustering on STevs's

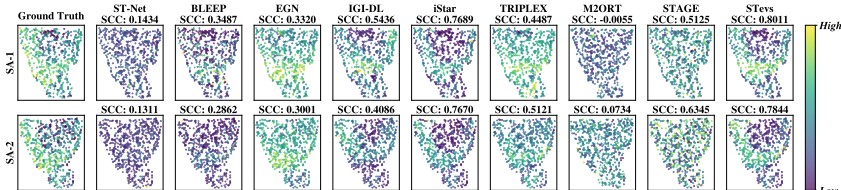

Figure 11: The results of intra-slice validation for the Dgkz gene on 2 datasets of anterior (with a 20% test set)

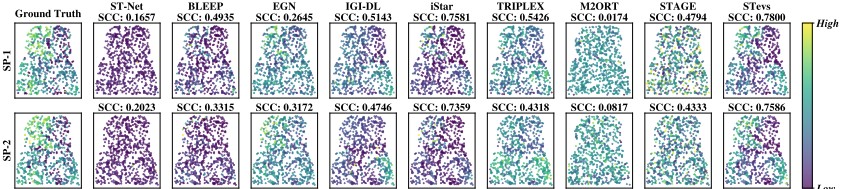

Figure 12: The results of intra-slice validation for the Dgkz gene on 2 datasets of posterior (with a 20% test set)

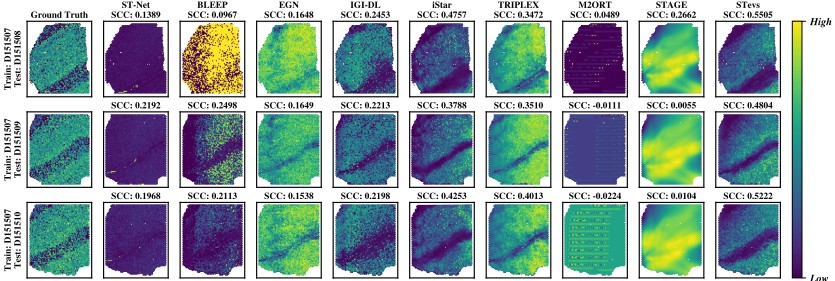

Figure 13: The cross-slice validation results of the OLFM1 gene on the other 3 slices of human1, with D151507 used as the training set

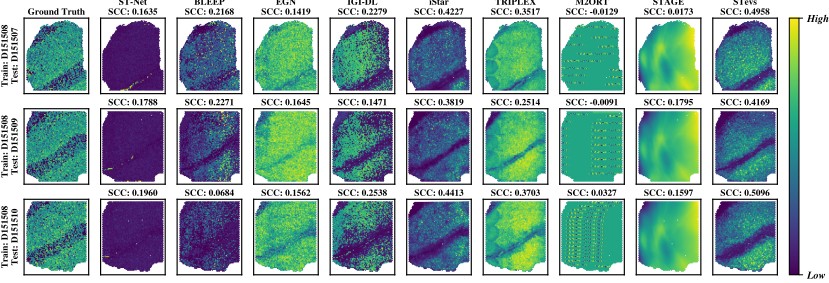

Figure 14: The cross-slice validation results of the OLFM1 gene on the other 3 slices of human1, with D151508 used as the training set

predictions not only significantly surpasses that of other advanced predictive models like iStar, but also consistently outperforms the baseline results from clustering on the original ground-truth RNA counts. This suggests that the predictions from STevs may serve a denoising functionEraslan et al. (2019), capturing the essential biological structures more clearly than the potentially noisy raw data, thereby enabling a more accurate recovery of the tissue's spatial domains.

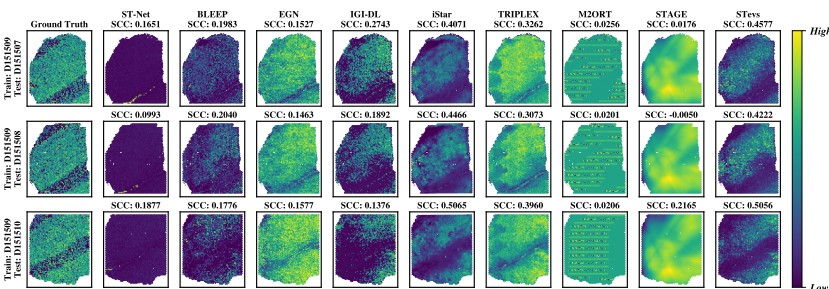

Figure 15: The cross-slice validation results of the OLFM1 gene on the other 3 slices of human1, with D151509 used as the training set

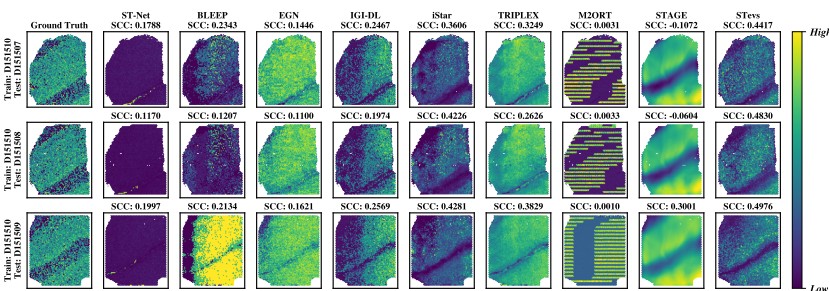

Figure 16: The cross-slice validation results of the OLFM1 gene on the other 3 slices of human1, with D151510 used as the training set

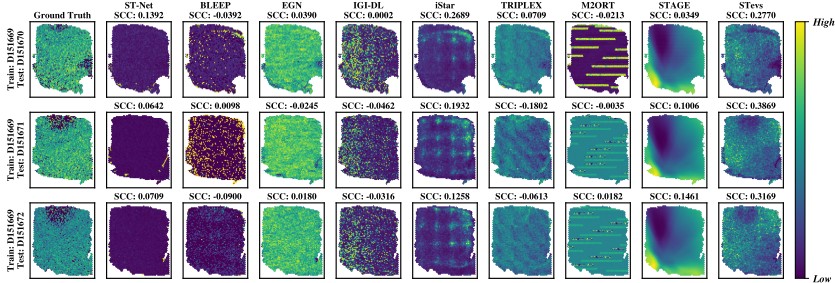

Figure 17: The cross-slice validation results of the OLFM1 gene on the other 3 slices of human2, with D151669 used as the training set

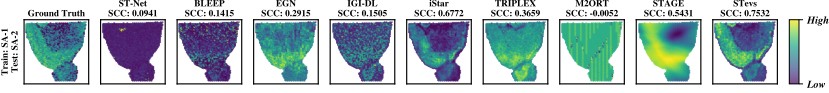

Figure 18: The cross-slice validation results of the Dgkz gene on the other 2 slices of anterior, with SA-1 used as the training set

# I  PARAMETER SENSITIVITY ANALYSIS

As shown in the Figure 20, we conducted a sensitivity analysis on three key hyperparameters: latent dimension, learning rate, and KLD loss weight. The model's performance is relatively sensitive to the choice of learning rate, while showing some, but not particularly high, sensitivity to the latent dimension and KLD loss weight. The parameters achieve optimal performance within a specific range.

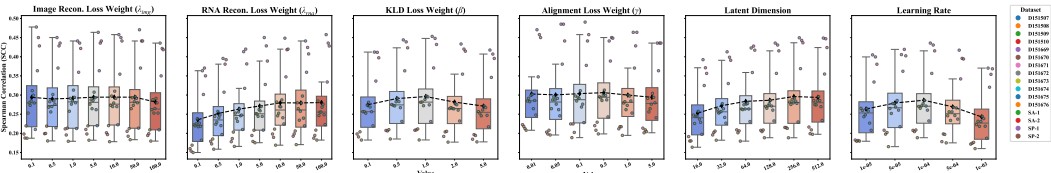

Figure 19: The cross-slice validation results of the Dgkz gene on the other 2 slices of posterior, with SP-1 used as the training set

Table 16: ARI Comparison across different methods and datasets

| | **DLPFC Dataset** | | | **10x Mouse Brain Dataset** | |
|---|---|---|---|---|---|
| **Method** | **Human 1** | **Human 2** | **Human 3** | **Sagittal-Anterior** | **Sagittal-Posterior** |
| RNA Counts | $0.139 \pm 0.032$ | $0.113 \pm 0.014$ | $0.176 \pm 0.009$ | $0.067 \pm 0.001$ | $0.053 \pm 0.001$ |
| STevs(Ours) | $\mathbf{0.246 \pm 0.040}$ | $\mathbf{0.238 \pm 0.069}$ | $\mathbf{0.238 \pm 0.051}$ | $\mathbf{0.280 \pm 0.002}$ | $\mathbf{0.295 \pm 0.018}$ |
| iStar | $0.214 \pm 0.037$ | $0.071 \pm 0.063$ | $0.172 \pm 0.075$ | $0.241 \pm 0.016$ | $0.221 \pm 0.006$ |
| TRIPLEX | $0.080 \pm 0.024$ | $0.018 \pm 0.016$ | $0.123 \pm 0.036$ | $0.098 \pm 0.004$ | $0.107 \pm 0.009$ |

Figure 20: Parameter Sensitivity Analysis

## J EXTENDED EXPERIMENT

### J.1 ROBUSTNESS EXPERIMENTS

#### J.1.1 METHODOLOGY

To comprehensively evaluate the robustness of our model, particularly its ability to handle color variations arising from different experimental batches or staining procedures, we designed and conducted a color augmentation simulation. We selected the H&E images from the D151673 and D151674 datasets and applied a **Spectral Blue Shift** transformation. This transformation is controlled by an intensity parameter, $\alpha$, which we varied from 0.1 to 1.0 in increments of 0.1, thereby generating a series of images with a progressively blueish hue. The key advantage of this method is its ability to induce a global spectral shift across the image without altering the microscopic cellular histology or macroscopic tissue structure, thus effectively simulating inter-slice color variations (i.e., batch effects).

The specific operation of this transformation on any given pixel color value, represented as $P = [R, G, B]^T$ (normalized to the range $[0, 1]$), is defined by the following mathematical formula:

$$P_{\text{shifted}} = \text{clip}_{[0,255]} \left( \begin{bmatrix} R \\ G \\ B \end{bmatrix} + 255 * \begin{bmatrix} -\alpha \\ -\alpha \\ +\alpha \end{bmatrix} \right) \tag{40}$$

where $P_{\text{shifted}}$ is the transformed pixel value and $\alpha$ is the parameter controlling the intensity of the color shift.

#### J.1.2 RESULTS

We tested the model on the images processed with varying intensities of the Spectral Blue Shift and recorded its performance. The experimental results, as illustrated in Figure 21, clearly show the trend of the model's performance as a function of the color shift intensity ($\alpha$). As observed in the figure, the performance of all models exhibited a decline with increasing spectral distortion. However, our proposed model demonstrated superior robustness. Compared to the baseline methods,

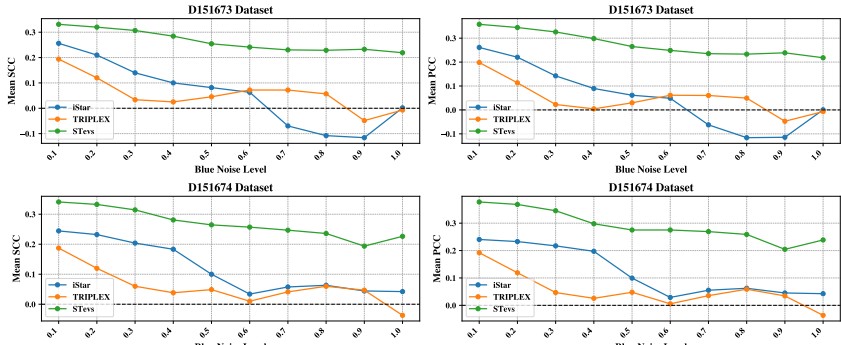

Figure 21: H&E stained images of the datasets used in the additional experiments. (Top row) Three HBC samples from a public dataset. (Bottom row) Three technical replicate samples of HSC from Ji et al. (2020).

our model's performance degradation curve was considerably flatter, with its advantages becoming more pronounced at higher intensity levels (e.g., $\alpha > 0.5$). This result strongly demonstrates that our model is insensitive to color variations in H&E images.

## J.2 EXTENDED DATASET EXPERIMENT

### J.2.1 DATASET

The first group of datasets consists of three Human Breast Cancer (HBC) Wu et al. (2021) samples, sourced from a publicly available Visium dataset. Breast cancer serves as a classic model for studying the heterogeneity of the Tumor Microenvironment (TME), as its tissue sections contain multiple cell types—including tumor, stromal, and infiltrating immune cells—that present a complex spatial architecture, making it highly suitable for evaluating the foundational generalization performance of a model. The second group is from a study on Human Squamous Cell Carcinoma (HSC) published by Ji et al. (2020), which is renowned for its high-quality multimodal data. From this, we selected three technical replicate sections from Patient 10. This provides an ideal validation scenario to rigorously test our model's stability and consistency when processing technical replicates from the same source tissue. The third group is derived from a MISAR-seq (Microfluidic Indexing-based Spatial Assay for Transposase-Accessible Chromatin and RNA-sequencing) Jiang et al. (2023) dataset, chosen to evaluate the model's capability in handling complex spatiotemporal and multi-omics data. This advanced dataset simultaneously provides spatial transcriptomics and spatial chromatin accessibility information from the same tissue section. We utilized data from different individuals at distinct developmental time points (E15 and E18) to challenge the model's robustness against biological variability. During the data preprocessing stage, we performed Spatially Variable Gene (SVG) filtering on each dataset. For the HBC dataset, a final set of 851 high-confidence SVGs was retained for subsequent experiments. Similarly, for the HSC dataset, we obtained 1483 SVGs. For the MISAR-seq dataset, a total of 678 SVGs were selected for subsequent experiments. The style of the sections for all datasets is illustrated in Figure 22.

Table 17: Intra-slice Performance comparison of different models on the HBC, HSC, and MISAR datasets.

| Method | HBC | | | HSC | | | MISAR | | |
|---|---|---|---|---|---|---|---|---|---|
| | MSE ↓ | PCC ↑ | SCC ↑ | MSE ↓ | PCC ↑ | SCC ↑ | MSE ↓ | PCC ↑ | SCC ↑ |
| IGI-DL | $0.8055 \pm 0.4531$ | $0.1269 \pm 0.0957$ | $0.1284 \pm 0.0761$ | $0.7310 \pm 0.8815$ | $0.1724 \pm 0.0674$ | $0.1349 \pm 0.0675$ | $0.9542 \pm 0.2588$ | $0.1150 \pm 0.0880$ | $0.1090 \pm 0.0813$ |
| iStar | $0.3804 \pm 0.1322$ | $0.3001 \pm 0.0652$ | $0.2441 \pm 0.0680$ | $0.8981 \pm 0.4588$ | $0.5789 \pm 0.0420$ | $0.4058 \pm 0.0354$ | $0.4958 \pm 0.5120$ | $0.3787 \pm 0.0450$ | $0.3595 \pm 0.0411$ |
| TRIPLEX | $0.2100 \pm 0.0216$ | $0.2390 \pm 0.0720$ | $0.2290 \pm 0.0701$ | $0.3070 \pm 0.0901$ | $0.3597 \pm 0.0526$ | $0.2884 \pm 0.0251$ | $0.5982 \pm 0.1105$ | $0.2591 \pm 0.0615$ | $0.2513 \pm 0.0588$ |
| **STevs (Ours)** | $\mathbf{0.1559 \pm 0.0162}$ | $\mathbf{0.3802 \pm 0.0672}$ | $\mathbf{0.3125 \pm 0.0598}$ | $\mathbf{0.1855 \pm 0.0487}$ | $\mathbf{0.5951 \pm 0.0313}$ | $\mathbf{0.4350 \pm 0.0374}$ | $\mathbf{0.3988 \pm 0.0415}$ | $\mathbf{0.3986 \pm 0.0391}$ | $\mathbf{0.3866 \pm 0.0352}$ |

### J.2.2 RESULTS

The detailed experimental results are presented in Table 17 (for the intra-slice task) and Table 18 (for the cross-slice task), demonstrating the superior performance of our model. In both prediction

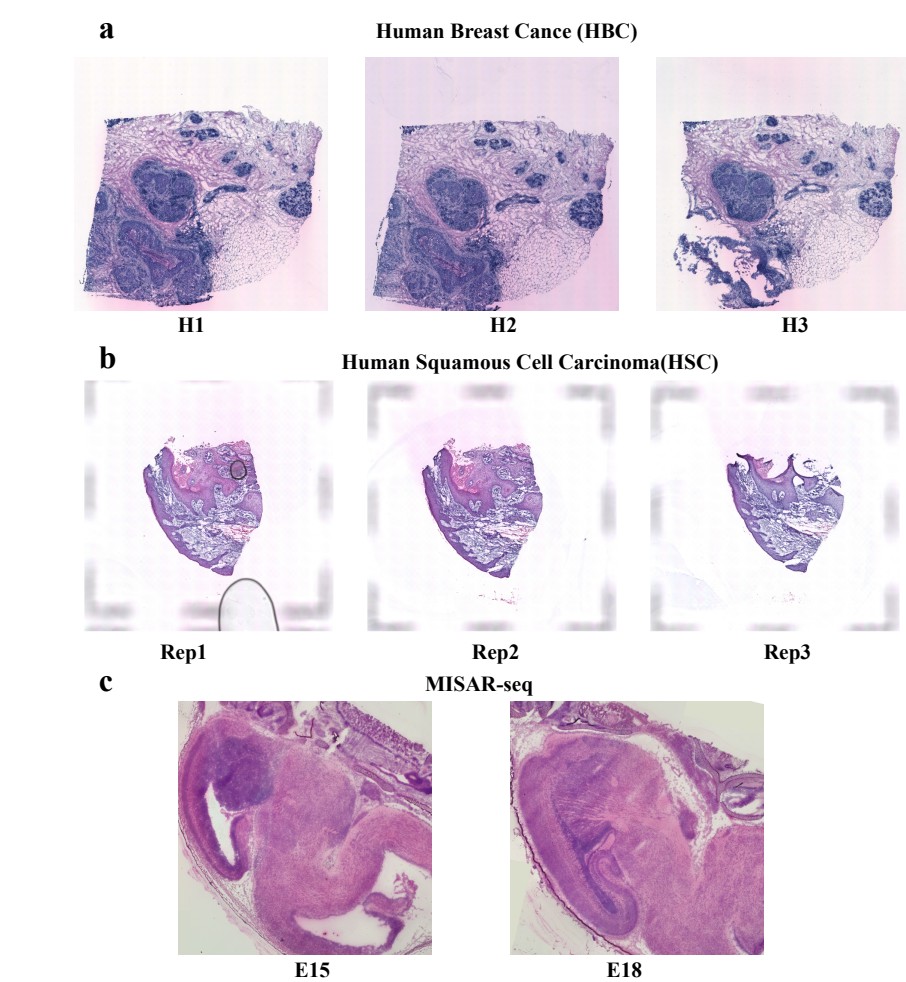

Figure 22: H&E stained images of the datasets used in the additional experiments. (Top row) Three HBC samples from a public dataset. (Bottom row) Three technical replicate samples of HSC from Ji et al. (2020).

Table 18: Cross-slice Performance comparison of different models on the HBC, HSC, and MISAR datasets.

| Method | HBC | | | HSC | | | MISAR | | |
|---|---|---|---|---|---|---|---|---|---|
| | MSE ↓ | PCC ↑ | SCC ↑ | MSE ↓ | PCC ↑ | SCC ↑ | MSE ↓ | PCC ↑ | SCC ↑ |
| IGI-DL | $1.3441 \pm 0.1693$ | $0.0899 \pm 0.0871$ | $0.0901 \pm 0.0908$ | $1.4990 \pm 0.1542$ | $0.1280 \pm 0.0629$ | $0.1147 \pm 0.0543$ | $1.6567 \pm 0.2289$ | $0.1287 \pm 0.0347$ | $0.1061 \pm 0.0321$ |
| iStar | $0.3977 \pm 0.0836$ | $0.2731 \pm 0.0221$ | $0.2013 \pm 0.0304$ | $0.9479 \pm 0.3782$ | $0.4849 \pm 0.0347$ | $0.3700 \pm 0.0176$ | $0.6946 \pm 0.0807$ | $0.1602 \pm 0.0145$ | $0.1521 \pm 0.0135$ |
| TRIPLEX | $0.1999 \pm 0.0272$ | $0.2577 \pm 0.0222$ | $0.2393 \pm 0.0252$ | $0.3256 \pm 0.0809$ | $0.3141 \pm 0.0243$ | $0.2626 \pm 0.0166$ | $0.9548 \pm 0.1671$ | $0.1096 \pm 0.0225$ | $0.1063 \pm 0.0188$ |
| STevs (Ours) | $\mathbf{0.1619} \pm 0.0092$ | $\mathbf{0.3640} \pm 0.0333$ | $\mathbf{0.2939} \pm 0.0260$ | $\mathbf{0.2719} \pm 0.0195$ | $\mathbf{0.5443} \pm 0.0338$ | $\mathbf{0.4039} \pm 0.0237$ | $\mathbf{0.5699} \pm 0.0553$ | $\mathbf{0.2087} \pm 0.0132$ | $\mathbf{0.2136} \pm 0.0110$ |

settings, STevs consistently outperforms all baseline methods across the HBC, HSC, and MISAR datasets, achieving the lowest MSE and the highest PCC and SCC correlations.

## K    ADDITIONAL EXPERIMENTAL DETAILS

### K.1    RUNNING TIME

We evaluated the computational efficiency of STevs against all baseline models. As detailed in the training time tables in the Appendix, STevs demonstrates excellent computational performance. In both intra-slice and cross-slice settings, the training time for STevs is significantly lower than that

of other high-performance Transformer-based models, such as iStar and M2ORT, and is one to two orders of magnitude faster than STAGE. This indicates that STevs maintains low computational overhead while achieving state-of-the-art predictive performance, showcasing an excellent balance between efficiency and accuracy.

### K.2 PARAMETER SETTINGS FOR OTHER METHODS

To ensure a fair comparison with baseline models and to facilitate the reproducibility of our results, we conducted a comprehensive hyperparameter search for all baselines. As detailed in the parameter search table in the Appendix, we defined a search space for the key hyperparameters of each model and determined the optimal parameter combination for each dataset independently. This targeted tuning strategy ensures that every baseline model was performing at or near its optimal state for comparison against STevs, thereby validating the rigor of our experimental evaluation.

Table 19: Intra-slice Training Time(s). Our experiments were conducted on a high-performance server equipped with four NVIDIA A100 GPUs (80GB of VRAM each), dual Intel(R) Xeon(R) Gold 6267C CPUs, and 1.5TB of system memory. The runtimes reported in the table are for running the model on a single GPU.

| Dataset | ST-Net | BLEEP | EGN | IGI-DL | iStar | TRIPLEX | M2ORT | STAGE | STevs |
|---|---|---|---|---|---|---|---|---|---|
| D151507 | 816.66 | 710.66 | 723.82 | 46.42 | 4263.32 | 329.58 | 3273.99 | 19840.76 | 315.07 |
| D151508 | 632.69 | 670.33 | 776.64 | 45.91 | 5057.34 | 311.76 | 3854.78 | 18280.31 | 349.79 |
| D151509 | 677.69 | 629.27 | 973.14 | 44.98 | 4784.25 | 368.39 | 3470.88 | 23201.72 | 354.20 |
| D151510 | 694.55 | 632.45 | 917.32 | 38.12 | 5266.73 | 328.05 | 3575.34 | 17053.08 | 332.63 |
| D151669 | 588.12 | 694.86 | 925.39 | 38.45 | 3457.32 | 269.61 | 3583.80 | 16024.78 | 288.73 |
| D151670 | 576.13 | 530.80 | 796.37 | 44.07 | 3211.23 | 249.29 | 3950.09 | 14621.26 | 238.13 |
| D151671 | 676.64 | 641.43 | 799.86 | 41.83 | 3415.35 | 298.14 | 3870.76 | 19408.45 | 272.04 |
| D151672 | 691.75 | 499.68 | 813.08 | 42.16 | 3454.13 | 280.87 | 3661.85 | 24743.11 | 277.08 |
| D151673 | 680.40 | 602.45 | 904.72 | 49.36 | 4236.42 | 283.91 | 3270.12 | 17625.90 | 268.60 |
| D151674 | 886.31 | 656.57 | 902.83 | 52.55 | 5283.45 | 279.64 | 3584.43 | 21101.03 | 258.41 |
| D151675 | 690.43 | 481.23 | 946.43 | 47.80 | 3436.56 | 421.84 | 3289.74 | 20454.75 | 246.32 |
| D151676 | 647.71 | 597.35 | 925.94 | 56.38 | 4203.35 | 274.28 | 3611.79 | 17363.65 | 302.48 |
| SA-1 | 639.54 | 514.74 | 878.00 | 53.71 | 4201.45 | 236.85 | 3467.52 | 19567.36 | 196.62 |
| SA-2 | 775.66 | 687.63 | 816.49 | 54.96 | 2376.21 | 249.96 | 3359.48 | 21511.63 | 198.71 |
| SP-1 | 851.26 | 444.32 | 961.98 | 56.53 | 3201.29 | 266.62 | 3690.96 | 24367.12 | 260.48 |
| SP-2 | 661.83 | 442.03 | 920.32 | 53.27 | 3815.53 | 266.39 | 3879.65 | 26614.65 | 272.23 |

Table 20: Cross-slice Training Time(s). Our experiments were conducted on a high-performance server equipped with four NVIDIA A100 GPUs (80GB of VRAM each), dual Intel(R) Xeon(R) Gold 6267C CPUs, and 1.5TB of system memory. The runtimes reported in the table are for running the model on a single GPU.

| Dataset | ST-Net | BLEEP | EGN | IGI-DL | iStar | TRIPLEX | M2ORT | STAGE | STevs |
|---|---|---|---|---|---|---|---|---|---|
| D151507 | 720.07 | 302.15 | 870.61 | 51.40 | 5813.85 | 715.52 | 4123.56 | 18426.30 | 413.85 |
| D151508 | 870.90 | 360.68 | 978.36 | 51.77 | 6344.25 | 672.18 | 4087.91 | 27341.57 | 494.83 |
| D151509 | 799.45 | 354.93 | 1385.73 | 46.03 | 6427.45 | 681.49 | 4210.34 | 21036.94 | 597.39 |
| D151510 | 862.53 | 403.01 | 1112.65 | 42.23 | 5132.24 | 795.83 | 3989.45 | 19508.30 | 508.69 |
| D151669 | 748.22 | 219.61 | 863.06 | 48.53 | 5383.45 | 655.76 | 4056.22 | 23193.66 | 598.03 |
| D151670 | 1024.44 | 282.19 | 857.38 | 57.46 | 6642.34 | 622.20 | 4188.76 | 21346.31 | 569.06 |
| D151671 | 792.82 | 313.80 | 968.52 | 57.28 | 5632.64 | 743.64 | 3998.11 | 22938.56 | 470.93 |
| D151672 | 981.05 | 310.15 | 928.61 | 56.01 | 4330.84 | 895.64 | 4065.99 | 17814.49 | 500.48 |
| D151673 | 899.40 | 309.10 | 1047.13 | 48.23 | 5245.45 | 839.44 | 4176.54 | 26316.95 | 425.79 |
| D151674 | 777.04 | 509.31 | 1126.61 | 47.37 | 5246.56 | 787.96 | 3954.88 | 24910.94 | 454.83 |
| D151675 | 841.35 | 406.07 | 1220.79 | 47.41 | 4256.57 | 786.02 | 4022.65 | 24798.90 | 556.87 |
| D151676 | 788.38 | 367.95 | 1095.60 | 48.63 | 6423.63 | 746.98 | 4101.99 | 28050.22 | 544.18 |
| SA-1 | 730.78 | 309.12 | 914.99 | 54.55 | 5356.29 | 472.68 | 4005.43 | 18317.55 | 454.80 |
| SA-2 | 915.74 | 359.40 | 1016.93 | 56.32 | 4356.43 | 493.94 | 3978.22 | 22109.28 | 284.63 |
| SP-1 | 901.73 | 371.56 | 1228.21 | 95.84 | 5485.25 | 651.50 | 4011.67 | 27389.33 | 518.49 |
| SP-2 | 935.85 | 338.43 | 1224.59 | 61.14 | 4352.24 | 653.60 | 4155.88 | 28165.83 | 527.08 |

Table 21: Baseline Parameter Search Results

| Model | Hyperparameter | Search Range | D151507 | D151508 | D151509 | D151510 | D151669 | D151670 | D151671 | D151672 | D151673 | D151674 | D151675 | D151676 | SA-1 | SA-2 | SP-1 | SP-2 |
|---|---|---|---|---|---|---|---|---|---|---|---|---|---|---|---|---|---|---|
| ST-Net | learning_rate | [1e-4, 5e-4, 1e-3] | 5e-04 | 1e-03 | 5e-04 | 5e-04 | 1e-04 | 5e-04 | 5e-04 | 1e-03 | 5e-04 | 5e-04 | 5e-04 | 5e-04 | 1e-03 | 5e-04 | 1e-03 | 1e-03 |
| | l2_reg | [0.001, 0.005, 0.01] | 0.005 | 0.005 | 0.001 | 0.005 | 0.005 | 0.01 | 0.005 | 0.01 | 0.005 | 0.005 | 0.005 | 0.005 | 0.001 | 0.001 | 0.005 | 0.001 |
| BLEEP | lr | [1e-4, 5e-4, 1e-3] | 1e-03 | 1e-03 | 1e-03 | 1e-03 | 1e-03 | 1e-03 | 1e-03 | 1e-03 | 1e-03 | 1e-03 | 1e-03 | 1e-03 | 1e-03 | 1e-03 | 1e-03 | 1e-03 |
| | hidden_dim | [128, 256, 512] | 256 | 256 | 256 | 512 | 128 | 256 | 256 | 256 | 512 | 256 | 256 | 256 | 512 | 256 | 512 | 512 |
| | lambda | [0.5, 1, 2] | 1 | 1 | 1 | 0.5 | 1 | 2 | 1 | 1 | 1 | 2 | 1 | 1 | 0.5 | 1 | 0.5 | 0.5 |
| EGN | lr | [1e-4, 5e-4, 1e-3] | 5e-04 | 5e-04 | 5e-04 | 5e-04 | 5e-04 | 5e-04 | 5e-04 | 5e-04 | 5e-04 | 5e-04 | 5e-04 | 5e-04 | 1e-03 | 1e-03 | 1e-03 | 1e-03 |
| | hidden_dim | [64, 128, 256] | 128 | 128 | 128 | 64 | 128 | 128 | 128 | 128 | 128 | 128 | 256 | 128 | 256 | 256 | 256 | 256 |
| | num_layers | [2, 3, 4] | 3 | 2 | 3 | 3 | 4 | 3 | 3 | 3 | 3 | 2 | 3 | 3 | 4 | 3 | 4 | 4 |
| | dropout | [0.3, 0.5, 0.7] | 0.5 | 0.5 | 0.5 | 0.5 | 0.3 | 0.5 | 0.7 | 0.5 | 0.5 | 0.5 | 0.5 | 0.5 | 0.3 | 0.5 | 0.3 | 0.3 |
| | lam | [0.1, 0.5, 1.0] | 0.5 | 0.5 | 0.5 | 1.0 | 0.5 | 0.5 | 0.5 | 0.1 | 0.5 | 0.5 | 0.5 | 0.5 | 1 | 1 | 1 | 1 |
| IGI-DL | lr | [1e-4, 5e-4, 1e-3] | 1e-04 | 1e-04 | 1e-04 | 1e-04 | 1e-04 | 1e-04 | 1e-04 | 1e-04 | 1e-04 | 1e-04 | 1e-04 | 1e-04 | 5e-04 | 5e-04 | 5e-04 | 5e-04 |
| | gat_hidden_dim | [128, 256, 512] | 256 | 256 | 256 | 256 | 256 | 256 | 256 | 256 | 256 | 256 | 256 | 256 | 128 | 128 | 128 | 128 |
| | gat_layer_num | [2, 3, 4] | 4 | 4 | 4 | 4 | 3 | 3 | 3 | 3 | 3 | 3 | 3 | 3 | 2 | 2 | 2 | 2 |
| | gat_dropout | [0.1, 0.2, 0.3] | 0.2 | 0.1 | 0.2 | 0.2 | 0.2 | 0.3 | 0.2 | 0.2 | 0.1 | 0.2 | 0.2 | 0.2 | 0.1 | 0.2 | 0.1 | 0.1 |
| iStar | lr | [1e-4, 5e-4, 1e-3] | 1e-04 | 5e-04 | 1e-03 | 5e-04 | 5e-04 | 5e-04 | 5e-04 | 5e-04 | 5e-04 | 1e-03 | 1e-03 | 1e-03 | 1e-03 | 1e-03 | 1e-03 | 1e-03 |
| | weight_decay | [1e-5, 1e-4, 1e-3] | 1e-04 | 1e-04 | 1e-04 | 1e-05 | 1e-04 | 1e-04 | 1e-04 | 1e-04 | 1e-03 | 1e-04 | 1e-04 | 1e-04 | 1e-05 | 1e-04 | 1e-05 | 1e-05 |
| TRIPLEX | learning_rate | [1e-4, 5e-4, 1e-3] | 1e-04 | 1e-04 | 1e-04 | 1e-04 | 1e-04 | 1e-04 | 1e-04 | 1e-04 | 5e-04 | 5e-04 | 5e-04 | 5e-04 | 1e-04 | 1e-04 | 1e-04 | 1e-04 |
| | n_hidden | [64, 128, 256] | 128 | 128 | 128 | 128 | 128 | 128 | 128 | 128 | 128 | 128 | 128 | 128 | 128 | 128 | 128 | 128 |
| | n_layers | [2, 3, 4] | 3 | 3 | 3 | 3 | 3 | 4 | 3 | 3 | 4 | 3 | 3 | 3 | 4 | 4 | 4 | 4 |
| | dropout | [0.1, 0.3, 0.5] | 0.3 | 0.3 | 0.1 | 0.3 | 0.3 | 0.3 | 0.3 | 0.5 | 0.3 | 0.3 | 0.3 | 0.1 | 0.5 | 0.3 | 0.5 | 0.5 |
| | weight_decay | [1e-5, 1e-4, 1e-3] | 1e-04 | 1e-04 | 1e-04 | 1e-04 | 1e-04 | 1e-04 | 1e-04 | 1e-04 | 1e-04 | 1e-04 | 1e-04 | 1e-04 | 1e-04 | 1e-04 | 1e-04 | 1e-04 |
| | alpha | [0.1, 0.5, 1.0] | 0.5 | 0.5 | 0.5 | 0.5 | 0.5 | 0.5 | 0.5 | 0.5 | 0.5 | 0.5 | 0.5 | 0.5 | 1 | 1 | 1 | 1 |
| M2ORT | lr | [1e-4, 5e-4, 1e-3] | 1e-04 | 1e-04 | 1e-04 | 1e-04 | 1e-04 | 1e-04 | 1e-04 | 1e-04 | 1e-04 | 1e-04 | 1e-04 | 1e-04 | 1e-04 | 1e-04 | 1e-04 | 1e-04 |
| | weight_decay | [1e-5, 1e-4, 1e-3] | 1e-04 | 1e-04 | 1e-04 | 1e-04 | 1e-04 | 1e-04 | 1e-04 | 1e-04 | 1e-04 | 1e-04 | 1e-04 | 1e-04 | 1e-05 | 1e-05 | 1e-05 | 1e-05 |
| STAGE | lr | [0.01, 0.005, 0.001] | 0.005 | 0.005 | 0.005 | 0.005 | 0.005 | 0.005 | 0.005 | 0.005 | 0.005 | 0.005 | 0.005 | 0.005 | 0.001 | 0.001 | 0.001 | 0.001 |
| | hidden_dims | [512, 256, 128] | 256 | 256 | 256 | 256 | 256 | 256 | 256 | 256 | 256 | 256 | 256 | 256 | 256 | 256 | 256 | 256 |
| | lambda_recon | [0.1, 1, 10] | 1 | 10 | 1 | 1 | 1 | 1 | 10 | 1 | 1 | 1 | 1 | 1 | 10 | 1 | 10 | 10 |
| | lambda_kl | [0.1, 1, 10] | 1 | 1 | 1 | 0.1 | 1 | 1 | 1 | 1 | 1 | 1 | 1 | 0.1 | 0.1 | 0.1 | 1 | 0.1 |
| | lambda_graph | [0.1, 1, 10] | 1 | 1 | 0.1 | 1 | 1 | 1 | 1 | 1 | 1 | 0.1 | 1 | 1 | 0.1 | 1 | 0.1 | 0.1 |

## L    THE USE OF LARGE LANGUAGE MODELS (LLMs)

During the preparation of this manuscript, we utilized Large Language Models (LLMs) as writing assistants. Specifically, we used Gemini Pro and DeepSeek to improve the grammar, clarity, and readability of the text. The models' role was strictly limited to rephrasing sentences for better flow and correcting typographical errors. The core scientific ideas, experimental design, analysis, and conclusions presented in this paper were conceived and developed entirely by the human authors. We have carefully reviewed and edited all model-generated text and take full responsibility for the final content of this paper, ensuring its scientific accuracy and originality.