# OpenReview forum: "Predicting Gene Expression in Spatially Resolved Transcriptomics Across Samples Through Probabilistic Fusion of Hierarchical Histology and Spatial Information"
_ICLR.cc/2026/Conference — ICLR 2026 Conference Withdrawn Submission_

### Official Review · Reviewer_2opC · 2025-10-28

**Soundness:** 2
**Presentation:** 2
**Contribution:** 2
**Rating:** 2
**Confidence:** 4

**Summary:**

This paper proposes STevs, a multimodal variational autoencoder that predicts gene expression from histology images in spatially resolved transcriptomics. The method uses a Swin Transformer for H&E image features and an MLP for spatial coordinates, which are fused probabilistically using a Product of Experts (PoE) mechanism that accounts for modality-specific uncertainty. The authors evaluate under two settings, including intra-slice and cross-slice.

**Strengths:**

1. The authors have clearly presented the core idea in Figure 2.
2. The task of predicting gene expression from H&E is high impact.

**Weaknesses:**

1. Many modules claimed to be novel are well established, e.g. NB loss for single cell, Hierarchical transformers (which Swin Transformer is a specialized case) are already adopted in iStar[1] and iSCALE[2].
2. The motivation for the image feature - spatial spot coordinate alignment is uninspiring. I'm not understanding why physical coordinates need to be aligned with image features,
3. The benchmark data split could be better performed, including using held-out regions rather than random sampling (Supplementary).  Spatial proximity has a high correlation with transcriptomic information
4. One major claim for the paper is cross-slide/patient generalization. I'm not seeing that being well addressed, because the samples from each study are from a very limited number of donors, and the data split is done at the slice level.


[1]Zhang, Daiwei, et al. "Inferring super-resolution tissue architecture by integrating spatial transcriptomics with histology." Nature biotechnology 42.9 (2024): 1372-1377.
[2]Schroeder, Amelia, et al. "Scaling up spatial transcriptomics for large-sized tissues: uncovering cellular-level tissue architecture beyond conventional platforms with iSCALE." Nature methods (2025): 1-12.

**Questions:**

1. For the ablation of the Swin Transformer, I'm not seeing details for the compared backbone on CNN and ViT. Are they pretrained? if yes, is the pretraining task & dataset aligned with swin transformer?
2. I understand one contribution of the paper is on constructing new benchmarks. Have you considered benchmarking with HEST[1]? which includes more diverse samples from a much larger number of donors. This can support your claims in my last comment in weakness.

[1] Jaume, Guillaume, et al. "Hest-1k: A dataset for spatial transcriptomics and histology image analysis." Advances in Neural Information Processing Systems 37 (2024): 53798-53833.

---

### Official Review · Reviewer_PzDZ · 2025-10-28

**Soundness:** 3
**Presentation:** 2
**Contribution:** 2
**Rating:** 4
**Confidence:** 4

**Summary:**

This paper proposes a new method for gene expression prediction from information fusion between HE images and spatial location.

**Strengths:**

The method is clearly presented. The task is important. The results look interesting.

**Weaknesses:**

Although the task is well-defined and the method is well-presented, I still have some questions about model design and model performance.

1. In Figure 1, why do we need to utilize image embeddings from UMAP? Does this mean the method input is 2-d UMAPs rather than original embeddings? UMAP is affected by randomness, and thus, the model stability will be affected.

2. In Figure 2, why do the authors want to retrain the image encoder? There are several Pathology Foundation Models that can give you better representation. Could we start from these models?

3. In Figure 2, the presentation of the spatial encoder is not clear. What is the unique contribution of using a spatial encoder? I think HE images have already contained a spatial difference.

4. The datasets lack diversity. The authors only select two datasets, and each dataset has replicates. HEST and STImage1k4M provide more datasets; therefore, the authors should consider including more.

5. There are also several new methods, such as STFlow and MERGE, in the benchmarking analysis.

**Questions:**

Please see the weaknesses.

**Details Of Ethics Concerns:**

NA.

---

### Official Review · Reviewer_XvLD · 2025-10-28

**Soundness:** 3
**Presentation:** 3
**Contribution:** 2
**Rating:** 4
**Confidence:** 3

**Summary:**

This paper studies the problem of spatial transcriptomics prediction from histology images, and introduces an VAE-based generative model framework that probabilistically fuses image features and spatial coordinates using a Product-of-Experts mechanism. The method is evaluated on two benchmark datasets under both intra-slice and cross-slice settings, consistently achieving state-of-the-art performance.

**Strengths:**

1. The studied problem is interesting and critical. Directly inferring gene expression from histology images can potentially help with clinical applications.
2. The experiments are comprehensive, including 8 baseline methods and complete ablation studies.

**Weaknesses:**

1. One of my concerns is the intra-slice evaluation setting used in this paper. It sounds like a spot-level imputation task and can be highly sensitive to the specific train–test split (the results are not evaluated using multiple random splits or random seeds). Moreover, it is much less practically important than cross-slice setting. I recommend that authors can explore additional downstream use cases, such as cross-slide retrieval, which can fully leverage the two-tower architecture of the proposed model.
2. There are some successful pathology foundation models, such as UNI [3] and Gigapath [4], and they have demonstrated promising results in predicting spatial transcriptomics [1,2]. I would suggest that applying these models as image encoders or consider them as baselines on HEST-Bench [1], which is a comprehensive benchmark and includes 10 cancer types.
3. At least two generative-model-based methods exist for this task [5,6], yet the manuscript does not discuss or compare with them.
4. For the single-modality inference results in Table 5, clarification is needed regarding whether the same pretrained multimodal model is used while masking one modality during inference. If that is the case, the comparison may be unfair, since the model has already trained from supervision from both modalities.


[1] Jaume, Guillaume, et al. "Hest-1k: A dataset for spatial transcriptomics and histology image analysis." *Advances in Neural Information Processing Systems* 37 (2024): 53798-53833.

[2] Chen, Jiawen, et al. "Stimage-1k4m: A histopathology image-gene expression dataset for spatial transcriptomics." Advances in Neural Information Processing Systems 37 (2024): 35796-35823.

[3] Chen, Richard J., et al. "Towards a general-purpose foundation model for computational pathology." *Nature medicine* 30.3 (2024): 850-862.

[4] Xu, Hanwen, et al. "A whole-slide foundation model for digital pathology from real-world data." *Nature* 630.8015 (2024): 181-188.

[5] Huang, Tinglin, et al. "Scalable Generation of Spatial Transcriptomics from Histology Images via Whole-Slide Flow Matching." *ICML* 2025.

[6] Zhu, Sichen, et al. "Diffusion generative modeling for spatially resolved gene expression inference from histology images." *ICLR* 2025.

**Questions:**

See weakness.

---

### Official Review · Reviewer_JPUi · 2025-11-01

**Soundness:** 3
**Presentation:** 2
**Contribution:** 3
**Rating:** 6
**Confidence:** 4

**Summary:**

This paper introduces a multimodal VAE framework for predicting gene expression from H&E histology images in SRT. The method employs parallel encoders (Swin Transformer for images, MLP for spatial coordinates) fused via Product of Experts (PoE), with a latent alignment loss to promote cross-modal consistency. The authors demonstrate improved cross-slice generalization compared to existing methods on DLPFC and mouse brain datasets.

**Strengths:**

- The cross-slice evaluation protocol is the right benchmark for assessing true generalization capability, unlike standard intra-slice splits that inflate performance estimates artificially.
- PoE fusion is theoretically motivated for uncertainty-aware multimodal integration, properly weighting modalities based on inverse variance rather than naive concatenation.
- The latent alignment loss (Eq. 2) provides explicit regularization for domain-invariant representations, addressing batch effects that plague cross-sample predictions.
- Ablation studies systematically validate design choices including fusion mechanisms, encoder architectures, and the necessity of each component.
- Using negative binomial likelihood for count data is statistically appropriate given overdispersion in spatial transcriptomics, unlike MSE or Poisson assumptions.
- Single-modality inference experiments (Table 5) demonstrate robustness when one modality is unavailable, suggesting the learned representations are not brittle.

**Weaknesses:**

- The spatial encoder is just a simple MLP on (x,y) coordinates with no principled spatial structure modeling. This ignores spatial autocorrelation and doesn't scale to irregular geometries or 3D tissues.
- Latent alignment loss (L2 between μ_img and μ_spatial) is crude - it enforces identical means but ignores variance structure and doesn't guarantee semantic consistency, only distributional matching.
- Image reconstruction as auxiliary task lacks justification - why should histology reconstruction help gene prediction when many visual features (e.g., stromal patterns) may be transcriptionally irrelevant?
- Product of Experts assumes conditional independence of modalities given latent z, but image patches and spatial location are clearly dependent (tissue architecture creates spatially structured visual patterns).
- Cross-slice improvements are modest, and absolute correlations remain low questioning practical utility for unseen samples.
- Gene selection uses spatially variable genes (SVGs) which biases evaluation toward genes with strong spatial structure that are easier to predict from coordinates alone.
- No comparison to foundation models or self-supervised pretraining on large histology datasets, which might capture generalizable tissue features better than ImageNet initialization.
- The paper claims 2K+ genes but doesn't report genome-wide performance or how prediction quality varies across different gene expression ranges or biological pathways.
UMAP visualizations show batch mixing but don't quantify whether biological signal is preserved - mixing could occur by discarding biology rather than removing batch effects.
- Training uses KL annealing but no discussion of final β value or posterior collapse diagnostics, which are critical failure modes for VAEs.
- Method requires paired H&E and spatial transcriptomics for training on each organ type - unclear if brain-trained models transfer to other tissues or if retraining is needed per organ.
- Computational cost and scalability not discussed - Swin Transformer on high-resolution patches for thousands of spots could be prohibitive for whole-slide inference.
- Statistical testing is absent - no confidence intervals or significance tests on performance differences, making it hard to assess if improvements are meaningful.
- The claim of virtual spatial transcriptomics oversells the method given PCC ~0.4 on cross-slice tasks - this is insufficient accuracy for replacing actual experiments in discovery settings.

**Questions:**

- Negative binomial decoder parameterization (μ, θ) isn't detailed - how is θ predicted? Is it gene-specific, spot-specific, or global? This matters for modeling overdispersion properly.
- How can the alignment loss, which forces the image and spatial representations to be similar, be reconciled with the probabilistic fusion mechanism? Isn't that fusion mechanism designed to optimally balance contributions based on their certainty, whereas forcing alignment overrides this?
- The KLD loss term constrains the combined latent representation. Was an analysis performed on the separate image and spatial representations for evidence of "representation collapse," where one of them becomes uninformative?
- The spatial encoder only learns the absolute position from spot coordinates. How does the model learn the relative spatial relationships between distant patches, a critical feature that graph-based methods explicitly capture?
- The gene decoder models over 2000 genes as separate and unrelated, given the learned latent representation. How does this strong assumption of separateness impact the model's ability to capture known groups of genes that work together?

---

### Note · Authors · 2025-11-14

**Comment:**

I have read and agree with the venue's withdrawal policy on behalf of myself and my co-authors.

**Withdrawal Confirmation:**

I have read and agree with the venue's withdrawal policy on behalf of myself and my co-authors.